# PARAMETER RELEASE AND KNOWLEDGE REUSE FOR CLASS-INCREMENTAL SEMANTIC SEGMENTATION

## ABSTRACT

Class-incremental semantic segmentation aims to progressively learn new classes while preserving previously acquired knowledge. This task becomes particularly challenging when prior training samples are unavailable due to data privacy or storage restrictions, resulting in catastrophic forgetting. To address this issue, knowledge distillation is widely adopted as a constraint by maximizing the similarity of representations between the current model (learning new classes) and the previous model (retaining old ones). However, knowledge distillation inherently preserves the old-knowledge distribution with minimal modification. This constraint limits the parameters available for learning new classes when substantial information from old classes is retained. Furthermore, the acquired old knowledge is often ignored to facilitate the learning of new knowledge, resulting in a waste of previously learned procedures. The above two problems result in the risk of class confusion and deviating from the performance of joint learning. Based on such analysis, we propose **D**istribution-based **K**nowledge **D**istillation (DKD) via a minimization–maximization distribution strategy. On the one hand, to alleviate the parameter competition between old and new knowledge, we minimize the distribution of old knowledge after releasing low-sensitivity parameters to old classes. On the other hand, to effectively utilize the valuable knowledge previously acquired, we maximize shared-knowledge distribution between the old and new knowledge after approximating the new knowledge distribution via Laplacian-based projection estimation. The proposed method achieves an excellent balance between stability and plasticity in nine diverse settings on Pascal VOC and ADE20K. Notably, its average performance approaches that of joint learning (upper bound) while effectively reducing class confusion. The source code is provided in the supplementary material and will be made publicly available upon acceptance.

## 1 INTRODUCTION

Recently, supervised semantic segmentation Xie et al. (2021); Jain et al. (2023); Liao & Kong (2025) has made significant progress, which typically requires closed-set datasets where all classes are obtained at once for manual annotation. However, in real-world applications, new classes continuously emerge and models trained on old data struggle to adapt to new class data. A naive approach is to retrain the model on a combined dataset of old and new classes, known as *joint training*, but this is time-consuming and the old data is often partially inaccessible due to privacy or storage restrictions Yang et al. (2023); Zhao et al. (2023); Li & Hoiem (2017); Baek et al. (2022). Therefore, class-incremental semantic segmentation (CISS) Cha et al. (2021b); Maracani et al. (2021); Zhang et al. (2022b) emerged with the goal of continuously learning new classes while mitigating forgetting old ones, even when old classes are unavailable. This task is essential in real-world scenarios such as autonomous driving, medical image analysis, and environmental monitoring.

In recent years, CISS methods have focused on catastrophic forgetting Cha et al. (2021b); Douillard et al. (2021b); Yuan & Zhao (2024) and background shift Cermelli et al. (2020); Park et al. (2024); Qiu et al. (2023), enabling models to retain previously learned knowledge while adapting to new data. To alleviate the aforementioned issues, some methods Yoon et al. (2017); Qin et al. (2021) attempt to dynamically expand the modules, but this strategy introduces an additional inference burden due to the extra parameters. Hence, most CISS methods Cha et al. (2021b); Douillard et al. (2021b); Baek et al. (2022); Shang et al. (2023); Wang et al. (2024) rely on static architectures, meaning the capacity

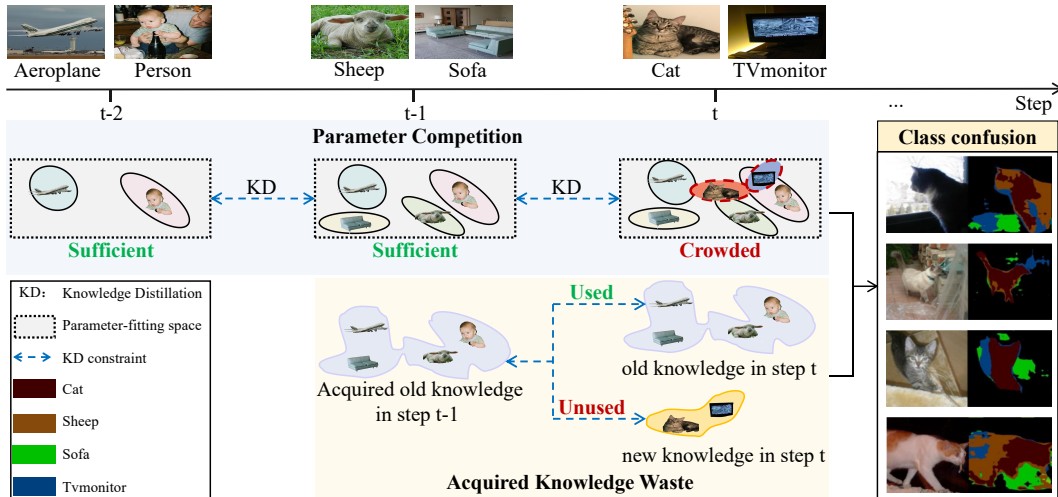

Figure 1: Problems and Analysis. KD, a widely adopted strategy, inherently preserves the old knowledge distribution with minimal change. However, as the number of new classes grows, this strategy aggravates parameter competition and causes the capacity for new-class distributions to become crowded. In addition, acquired old knowledge remains unused for step-wise guidance, neglecting its valuable role in supporting the learning of new knowledge.

of the overall parameter-fitting knowledge space is fixed. Pseudo-label based cross-entropy (CE) Cha et al. (2021b); Cermelli et al. (2020) and knowledge distillation (KD) Zhao et al. (2023); Wang et al. (2024); Yang et al. (2022); Shang et al. (2023); Baek et al. (2022); Cong et al. (2023) are typical strategies in CISS. Mainstream KD approaches used in static architectures can be broadly categorized into two types. The first performs multi-level feature-based knowledge distillation (MKD) by enforcing similarity between the frozen old model and the new model Zhao et al. (2023); Wang et al. (2024). The second, to alleviate guidance from potentially incorrect pixel-level information in the frozen old model, employs confidence map-based knowledge distillation (CKD), which retains high-confidence pixel information for feature distillation Yang et al. (2022); Shang et al. (2023). These mainstream KD approaches under a static parameter-fitting space hide a parameter competition: excessive parameter fitting for old knowledge hinders the learning of new classes, while overfitting to new knowledge leads to forgetting of old classes. When using KD to construct the similarity of the old representations during different steps, the fitting capacity for new knowledge becomes increasingly crowded as more classes are learned. This leads to a crowded distribution of new knowledge, as illustrated in the light-blue region of Fig. 1. Moreover, this constraint causes the valuable knowledge previously acquired to remain unused when learning new classes, as shown in the light-yellow region of Fig. 1. Consequently, the model struggles to balance stability and plasticity and becomes prone to class confusion, as illustrated in the right half of Fig. 1. This raises a critical question: **Can we release parameters and simultaneously reuse the acquired knowledge when learning new knowledge without introducing an additional inference burden?**

To address the above issue, we propose a **D**istribution-based **K**nowledge **D**istillation (DKD) via the minimization–maximization strategy. DKD releases previously occupied low-sensitivity parameters to alleviate competition between old and new knowledge. Additionally, Laplacian-based projection estimation produces two attention maps that identify reusable knowledge to guide the learning of new knowledge: (1) A position map that represents the spatial regions in which all categories coexist harmoniously, and (2) a confidence map that indicates the necessity of spatial reuse of old category information within the distribution of the new model. Subsequently, we maximize the shared knowledge distribution between old and new knowledge. As shown in Fig. 2(a-c), unlike prior KD methods, DKD emphasizes parameter release and knowledge reuse for the learning of new knowledge to mitigate parameter competition and reduce wasted knowledge. Experiments demonstrate that dynamic parameter release facilitates greater parameter differences between the incremental and initial steps (Fig. 2(d)). Furthermore, knowledge reuse in DKD not only mitigates catastrophic forgetting (as seen in the slower performance drop across incremental steps in the green curve of Fig. 2(e)), but also significantly enhances new class learning, achieving higher pixel-level classification accuracy with increasing incremental steps (Fig. 2(f)). Experiments on Pascal VOC

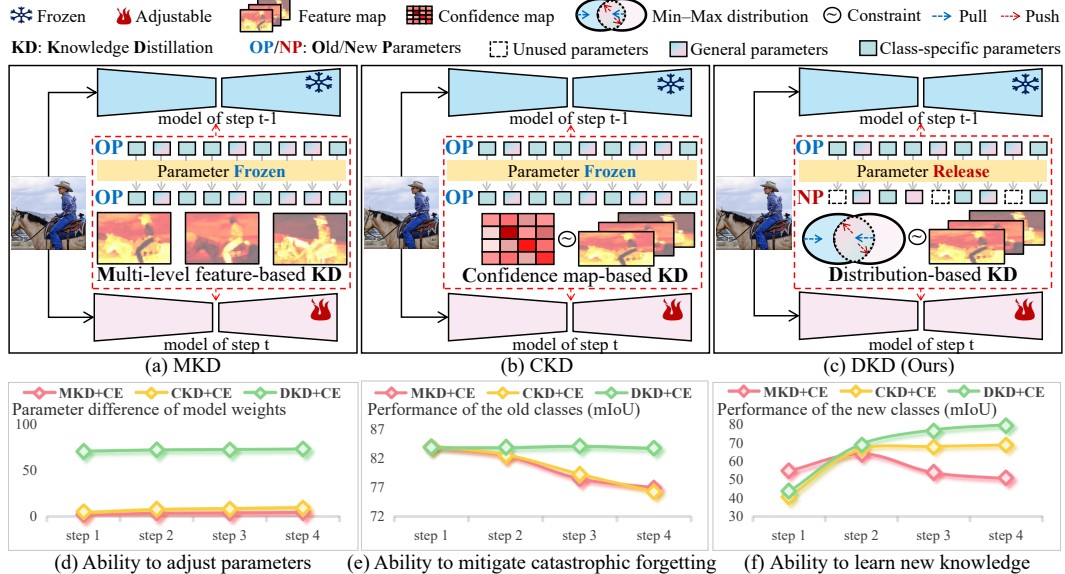

Figure 2: Comparison with existing KD methods. Unlike previous approaches that freeze the old model, our method dynamically adjusts knowledge distribution through parameter release and knowledge reuse (as shown in a–c). This enables dynamic parameter adjustment (as shown in d), mitigates catastrophic forgetting (as shown in e), and enhances new class learning (as shown in f).

2012 and ADE20K datasets demonstrate the effectiveness of our approach. Our main contributions are summarized as follows:

- We show that mainstream KD for CISS overlooks *parameter competition* and *underutilization of acquired knowledge*, motivating our Distribution-based Knowledge Distillation (DKD) with a **minimization–maximization** distribution strategy.

- To mitigate parameter competition, we **minimize** the distribution of old knowledge after releasing low-sensitivity occupied parameters; to reduce the underutilization of acquired knowledge, we reuse the acquired old knowledge to guide new class learning and **maximize** the shared knowledge distribution. These two components constitute DKD.

- Theoretical analysis and extensive experiments validate our approach, delivering state-of-the-art average performance and results close to the joint-training (upper bound).

## 2 RELATED WORK

In Class-Incremental Semantic Segmentation (CISS), the model is required to continuously learn new classes for semantic segmentation while preserving the knowledge of previously learned classes, addressing the challenges of catastrophic forgetting and background shift. Current research on CISS can be broadly divided into three main types of methods: 1) Replay-based approaches Cha et al. (2021b); Maracani et al. (2021); Rebuffi et al. (2017); Chen et al. (2025); Yu et al. (2024); Oh et al. (2022), which mitigate forgetting by storing or generating past data for rehearsal. Recall Maracani et al. (2021) leverages generative adversarial networks and web-crawled data to synthesize replayable samples of previously learned classes for continual learning. To avoid the burden of storing old data, TiKP Yu et al. (2024) employs text-to-image generation to replay samples of previously learned classes. 2) Dynamic architecture-based approaches Yoon et al. (2017); Qin et al. (2021); Aljundi et al. (2017); Yan et al. (2021), which expand the network structure to accommodate new classes. BNS Qin et al. (2021) dynamically constructs task-specific networks to mitigate catastrophic forgetting while facilitating knowledge transfer across tasks. EG Aljundi et al. (2017) progressively introduces task-specific experts as the number of tasks increases, enabling dynamic expert selection during inference. 3) Regularization-based approaches preserve previously learned knowledge by constraining representations. To address background shift, MIB Cermelli et al. (2020) remodels the background in the ground truth based on the output of the previous steps. MicroSeg Zhang et al. (2022b) introduces mask proposals to refine pseudo-labels. Building on the effectiveness of knowledge distillation to alleviate catastrophic forgetting Yang et al. (2019); Heo et al. (2019);

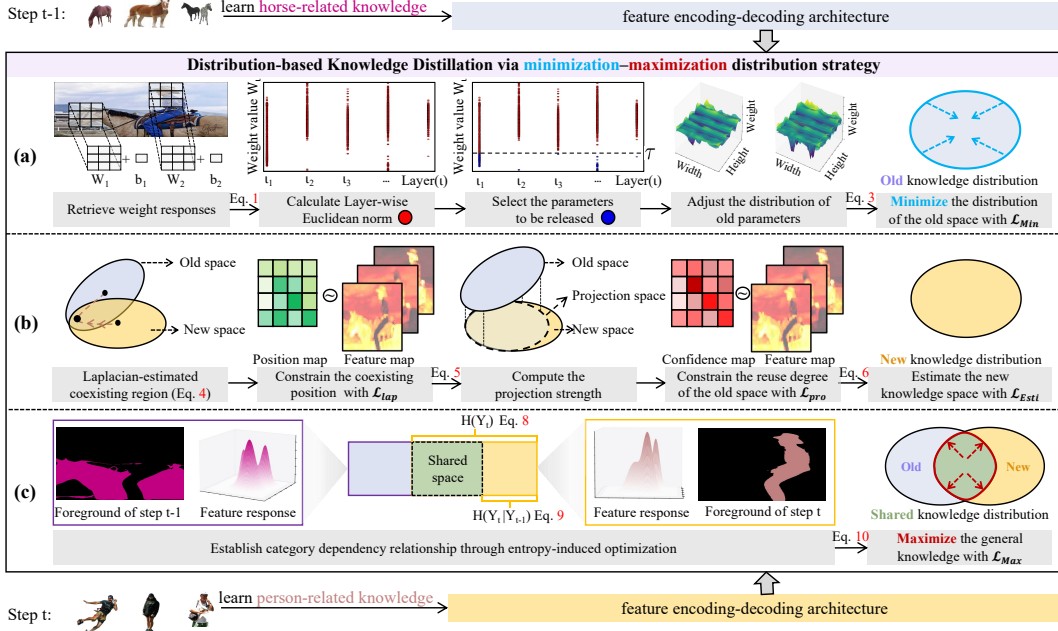

Figure 3: Overview of the proposed Distribution-based Knowledge Distillation (DKD) via a minimization–maximization distribution strategy. Our strategy introduces distributional constraints between old and new knowledge—minimizing the old knowledge distribution via $\mathcal{L}_{Min}$, while maximizing the shared knowledge through $\mathcal{L}_{Max}$ after estimating the new knowledge distribution with $\mathcal{L}_{Esti}$. This approach achieves parameter release (via $\mathcal{L}_{Min}$) and knowledge reuse (via $\mathcal{L}_{Esti}$ and $\mathcal{L}_{Max}$).

Wang & Yoon (2021), IL Michieli & Zanuttigh (2019) leverages the outputs of the old model, typically with frozen parameters, to supervise the trainable new model by maximizing the similarity between intermediate representations and logits via distillation losses, a process known as multi-level feature map-based knowledge distillation (MKD). Methods such as Comformer Cermelli et al. (2023), PLOP Douillard et al. (2021b), and CLIS Zheng et al. (2021) adopt a confidence map-based knowledge distillation (CKD), where high-confidence regions from the old model are selected and retained at the pixel level to guide the distillation process. However, our method differs in that our goal is to mitigate parameter competition between old and new knowledge while promoting the reuse of acquired knowledge, thereby effectively balancing stability and plasticity in CISS.

## 3 METHOD

### 3.1 PROBLEM DEFINITION

In CISS, the learning process involves a sequence of incremental steps, denoted as t=1,2,3,..., T. Following prior works Cha et al. (2021b); Douillard et al. (2021b); Cermelli et al. (2020; 2023), we adopt the overlapped setting, where the ground truth only includes labels for the currently learned class $C_t$, while all other classes, both previously learned and those to be presented in future steps, are treated as background $C_0$. The CISS model $f_{t,\theta}$ in the t-th incremental learning step is parameterized by $\theta$, comprising a feature extractor and a classifier. Given an input image $x$, the predicted result is obtained as $y_t$, where the predicted category may belong to the old classes $C_{1:t-1}$, the new classes $C_t$, or the background class $C_0$.

### 3.2 DISTRIBUTION-BASED KNOWLEDGE DISTILLATION

**(a) Parameter-driven minimization of old knowledge distribution.** Rather than freezing the parameters $\theta_{t-1}$ of the old model, we dynamically update their utilization at each step to enable adaptive knowledge retention and effective parameter reuse. To achieve this, as shown in Fig. 3(a), we first retrieve the weight matrix $W_l$ for each layer $l$ in the model. Then we compute the layer-wise Euclidean norm (L2) of the weight matrix to assess its importance. For layer $l$, the weight matrix for a convolutional layer is $W_l^{\text{Conv}} \in \mathbb{R}^{C_{\text{out}} \times C_{\text{in}} \times H_k \times W_k}$, and for a fully connected layer, $W_l^{\text{FC}} \in \mathbb{R}^{C_{\text{out}} \times C_{\text{in}}}$. Here, $C_{\text{out}}$ and $C_{\text{in}}$ denote the output and input channels (e.g., 3 for RGB), and $H_k$ and $W_k$ represent

the height and width of the kernel. The calculation for each layer is as follows:

$$
\|W_l\|_2 = \begin{cases} \sqrt{\sum_{o=1}^{C_{\text{out}}} \sum_{i=1}^{C_{\text{in}}} \sum_{h=1}^{H_k} \sum_{w=1}^{W_k} (W_{l,o,i,h,w})^2}, & \text{if } W_l^{\text{Conv}} \text{ is a convolutional layer.} \\ \sqrt{\sum_{o=1}^{C_{\text{out}}} \sum_{i=1}^{C_{\text{in}}} (W_{l,o,i})^2}, & \text{if } W_l^{\text{FC}} \text{ is a fully connected layer.} \end{cases} \tag{1}
$$

After computing the Euclidean norm, we apply a pruning threshold $\tau = 0.1$ (with experimental analysis in Appendix C.5) to retain significant weights and prune those below the threshold. Specifically, if the norm of a filter or output unit is less than $\tau$, it is pruned, and its weights and bias are set to zero. We define a binary pruning mask $P_l$ for each layer, where $P_{l,i} = 1$ means the $i$-th unit in the $l$-th layer is kept, and $P_{l,i} = 0$ means it is pruned. The selection decision is expressed as $P_{l,i} = 1$ when $\|W_{l,i}\|_2 \geq \tau$, and $P_{l,i} = 0$ otherwise. Based on the resulting binary mask $P_l$, the weights $W_l$ and biases $B_l$ of the old model are updated according to $\hat{W}_l = W_l \odot P_l$ and $\hat{B}_l = B_l \odot P_l$, respectively. By adjusting the utilization of the old parameters in this way, the model maintains essential knowledge while freeing up capacity for new classes. For each pixel location $(h, w)$ associated with previously learned classes c $\in 1, ..., C_{t-1}$, the label $y_c^*(h, w)$ is updated to alleviate background shift:

$$
y_c^*(h, w) = \mathbb{1}[y_t(h, w) = c]\hat{y}_{t-1}^c(h, w), \tag{2}
$$

where $y_t(h, w)$ is the prediction of step $t$ and $\hat{y}_{t-1}$ represents the prediction after parameter release. $\mathbb{1}$ denotes an indicator function that outputs 1 if the condition inside the parentheses is satisfied, and 0 otherwise. To dynamically minimize the distribution of old classes in $y_t(h, w)$ during the current learning step $t$ after parameter release, an optimization loss is introduced:

$$
\mathcal{L}_{Min} = -\frac{1}{H \times W} \sum_{h=1}^{H} \sum_{w=1}^{W} y_c^*(h, w)y_t(h, w). \tag{3}
$$

Through this computation, the output distribution of the old classes in the current-step model is encouraged to closely match the output of the old model after parameter release. In this process, the output of the parameter-released old model serves as the bound for this optimization, enabling the current step model to dynamically minimize the usage of old-class parameters via the loss constraint. Further theoretical analysis is provided in Appendix A.1.

**(b) Laplacian-based projection estimation for constructing the new knowledge distribution.** To effectively leverage previously acquired knowledge when forming the new knowledge distribution, we calculate a position map and a confidence map to identify where old knowledge is reusable and how strongly it applies, and we constrain their utilization via the loss $\mathcal{L}_{Esti}$. We compute the second-order gradient to identify low-curvature regions and store them in the position map $\mathcal{P}_t(h, w)$, where the representations of old and new knowledge exhibit minimal difference and can coexist.

$$
\mathcal{P}_t(h, w) = \left(\frac{\partial^2}{\partial h^2} + \frac{\partial^2}{\partial w^2}\right) \left[y_c^*(h, w) \|f_t(h, w) - f_{t-1}(h, w)\|_2\right] \tag{4}
$$

Additionally, the confidence map $\mathcal{C}_t(h, w)$ is generated to quantify the projection strength of old knowledge along the direction of the new knowledge distribution. Given the current model feature $f_t(h, w)$ and the label $y_c^*(h, w)$ of Eq. 2, the confidence map is computed as:

$$
\mathcal{C}_t(h, w) = \frac{< y_c^*(h, w), f_t(h, w) >}{\|f_t(h, w)\|_2}. \tag{5}
$$

Here, $< \cdot, \cdot >$ denotes the dot product. A higher value of $\mathcal{C}_t(h, w)$ indicates a stronger projection between $y_c^*(h, w)$ and $f_t(h, w)$, suggesting a higher likelihood of reusing previous knowledge in that region. Finally, to enforce the alignment between the feature representation $f_t$ and the inferred coexisting region, we define the loss $\mathcal{L}_{lap}$ with the guidance of $\mathcal{P}(h, w)$. Besides, we leverage the confidence map to guide the $\mathcal{L}_{pro}$ loss to constrain the model to preserve old knowledge representations in high-confidence regions. As shown in Fig. 3(b), the overall loss of Laplacian-based projection estimation for the construction of a new knowledge distribution is defined as:

$$
\mathcal{L}_{Esti} = \overbrace{\left(1 - \frac{1}{H \times W} \sum_{h=1}^{H} \sum_{w=1}^{W} \frac{\mathcal{C}_t(h, w)y_c^*(h, w)}{\|f_t(h, w)\|_2}\right)}^{\mathcal{L}_{pro}} + \gamma \overbrace{\|f_t - \mathcal{P}_t\|_2}^{\mathcal{L}_{lap}}, \tag{6}
$$

where $\gamma = 1$ by default, with experiments in Section 4.3 and theoretical analysis in Appendix A.2.

**(c) Entropy-induced optimization of overlap between new and old knowledge distribution.** To maximize the shared knowledge between the old and new distributions, we propose a loss constraint $\mathcal{L}_{\text{Max}}$ based on information theory Ash (2012). The objective is to encourage the model to exhibit

low conditional entropy and high marginal entropy in its predictions. Given a batch of predicted probabilities $y^b_{(t,c)} \in \mathbb{R}$ for category $c \in \{1, \ldots, C_t\}$ and sample $b \in \{1, \ldots, B\}$, we define marginal probability $\overline{y}_{(t,c)}$ as the average prediction across the entire batch for class $c$:

$$\overline{y}_{(t,c)} = \frac{1}{B} \sum_{b=1}^{B} y^b_{(t,c)}. \tag{7}$$

The marginal entropy is then calculated as:

$$H(Y_t) = -\sum_{c=1}^{C_t} \overline{y}_{(t,c)} \log(\overline{y}_{(t,c)}), \tag{8}$$

where high marginal entropy $H(Y_t)$ reflects a well-balanced category distribution in the new feature space, helps to avoid overfitting to specific old or new categories and promotes diversity. The conditional entropy, measuring the uncertainty of new knowledge given the old, is defined as:

$$H(Y_t \mid Y_{t-1}) = -\frac{1}{B} \sum_{b=1}^{B} \sum_{c=1}^{C_t} y^b_{(t,c)} \log(y^b_{(t,c)}), \tag{9}$$

where low conditional entropy $H(Y_t \mid Y_{t-1})$ indicates that the new knowledge distribution establishes effective dependencies on the old knowledge, facilitating knowledge retention and constructive transfer of prior information to form class-specific knowledge of step $t$. Finally, as shown in Fig. 3(c), we define the $\mathcal{L}_{\text{Max}}$ loss to maximize the overlap between the old and new distributions as:

$$\mathcal{L}_{\text{Max}} = -\frac{H(Y_t) - H(Y_t \mid Y_{t-1})}{\log |C_{t-1}|}, \tag{10}$$

where $|C_{t-1}|$ is the number of old classes for normalization. This loss encourages more general knowledge between the old and new class distributions by maximizing shared knowledge representation while maintaining discriminative diversity across classes. Related theoretical analysis is provided in Appendix A.3.

The overall optimization objective of DKD is formulated as follows:

$$\mathcal{L}_{\text{total}} = \mathcal{L}_{\text{Min}} + \mathcal{L}_{\text{Esti}} + \mathcal{L}_{\text{Max}} + \mathcal{L}_{\text{CE}}, \tag{11}$$

where $\mathcal{L}_{\text{CE}}$ denotes the cross-entropy loss Zhang & Sabuncu (2018).

## 4 EXPERIMENTS

### 4.1 EXPERIMENT SETUPS

Following previous works Cermelli et al. (2020); Yang et al. (2023); Zhang et al. (2023); Park et al. (2024), we evaluate our method on Pascal VOC 2012 Everingham et al. (2010) and ADE20K Zhou et al. (2017). We use ViT-B/16 Dosovitskiy et al. (2021) pretrained on ImageNet Deng et al. (2009) as the backbone and a decoder with two transformer blocks at 512×512 resolution. Optimization is performed using SGD Ketkar (2017) with momentum 0.9 and weight decay $1 \times 10^{-5}$ for 64 epochs per step. The learning rate starts at $1 \times 10^{-3}$ and is reduced to $1 \times 10^{-4}$ (Pascal VOC) or $5 \times 10^{-4}$ (ADE20K) in incremental steps. Experiments are run on 6 RTX 3090 GPUs with an Intel Xeon Gold 6226R CPU using PyTorch. Performance is evaluated using mean Intersection over Union (mIoU).

### 4.2 COMPARISON WITH THE STATE-OF-THE-ARTS

**Quantitative analysis on Pascal VOC 2012 and ADE20K.** In addition to the widely explored incremental settings of 15-1, 19-1, and 15-5 in previous works, we further evaluate the effectiveness of our method in more challenging scenarios, specifically 10-1 and 2-2. These two settings involve a larger number of incremental steps, making them more representative of practical incremental learning scenarios. As shown in Tab. 1, our method achieves notable improvements over previous approaches. For example, under the 2-2 setting, our method achieves a $3.9\%$ improvement on the old classes and a $1.8\%$ gain on the new classes, leading to an overall improvement of $4.7\%$ in the combined class performance. When averaged across all five incremental settings, our method achieves a gain of $1.7\%$, which demonstrates its effectiveness and generalizability in various CISS settings. We further evaluate the effectiveness of our method on the ADE20K dataset under 4 different incremental settings, as shown in Tab. 2. The results demonstrate robustness in handling more complex and large-scale CISS tasks. The average performance closely matches that of joint training, which is considered the upper bound under this architecture in CISS.

**Qualitative analysis of forgetting resistance on old classes and plasticity on new classes.** As shown in Fig. 4(a), prior methods tend to misclassify parts of the class "horse" as "sheep", indicating partial forgetting of old knowledge. Besides, previous methods show unsatisfactory sensitivity to new

Table 1: Comparative experiments on VOC dataset Everingham et al. (2010). The optimal and suboptimal performance are respectively represented in **red** and **blue** bold. The † symbol indicates results reproduced following the publicly released code. Across five incremental settings, our method achieves the highest average mIoU, demonstrating robust learning ability.

| Method | Backbone | 10-1 (11 steps) | | | 2-2 (10 steps) | | | 15-1 (6 steps) | | | 19-1 (2 steps) | | | 15-5 (2 steps) | | | Average |
|---|---|---|---|---|---|---|---|---|---|---|---|---|---|---|---|---|---|
| | | Old | New | All | Old | New | All | Old | New | All | Old | New | All | Old | New | All | |
| Joint | ViT | 85.0 | 84.7 | 84.9 | 77.3 | 85.5 | 84.3 | 83.9 | 79.1 | 82.8 | 84.4 | 79.6 | 84.2 | 85.5 | 80.3 | 84.3 | 84.1 |
| MIB Cermelli et al. (2020) | Resnet101 | 12.3 | 13.1 | 12.7 | 41.1 | 23.4 | 25.9 | 34.2 | 13.5 | 29.3 | 71.4 | 23.6 | 69.1 | 76.4 | 50.0 | 70.1 | 41.4 |
| SDR Michieli & Zanuttigh (2021) | Resnet101 | 32.1 | 17.0 | 24.9 | 13.0 | 5.1 | 6.2 | 44.7 | 21.8 | 39.2 | 69.1 | 32.6 | 67.4 | 75.4 | 52.6 | 70.0 | 41.5 |
| PLOP Douillard et al. (2021a) | Resnet101 | 44.0 | 15.5 | 30.4 | 24.1 | 11.9 | 13.6 | 65.1 | 21.1 | 54.6 | 75.4 | 37.4 | 73.6 | 75.7 | 51.7 | 70.0 | 48.4 |
| SSUL Cha et al. (2021a) | Resnet101 | 74.0 | 53.2 | 64.1 | - | - | - | 78.4 | 49.0 | 71.4 | 77.8 | 49.8 | 76.5 | 78.4 | 55.8 | 73.0 | - |
| MicroSeg Zhang et al. (2022b) | Resnet101 | 77.2 | 57.2 | 67.7 | 60.0 | 50.9 | 52.2 | 81.3 | 52.5 | 74.4 | 79.3 | 62.9 | 78.5 | 82.0 | 59.2 | 76.6 | 69.9 |
| REMINDER Phan et al. (2022) | Resnet101 | - | - | - | - | - | - | 68.3 | 27.7 | 58.6 | 76.5 | 32.3 | 74.4 | 76.1 | 50.7 | 70.1 | 67.7 |
| RCIL Zhang et al. (2022a) | Resnet101 | 55.4 | 15.1 | 36.2 | 28.3 | 19.0 | 20.3 | 70.6 | 23.7 | 59.4 | 68.5 | 12.1 | 65.8 | 78.8 | 52.0 | 72.4 | 50.8 |
| EWF Xiao et al. (2023) | Resnet101 | 71.5 | 30.3 | 51.9 | - | - | - | 77.7 | 32.7 | 67.0 | 77.9 | 6.7 | 74.5 | - | - | - | - |
| LGKD Yang et al. (2023) | Resnet101 | - | - | - | - | - | - | 70.6 | 30.9 | 61.1 | 77.3 | 42.9 | 75.7 | 79.5 | 54.8 | 73.6 | - |
| IDEC Zhao et al. (2023) | ResNet101 | 70.7 | 46.3 | 59.1 | - | - | - | 77.0 | 36.5 | 67.4 | - | - | - | 78.0 | 51.8 | 71.8 | - |
| GSC Cong et al. (2023) | Resnet101 | 50.6 | 17.3 | 34.7 | - | - | - | 72.1 | 24.4 | 60.7 | 76.9 | 42.7 | 75.3 | 78.3 | 54.2 | 72.6 | - |
| CoMasTRe Gong et al. (2024) | ResNet101 | - | - | - | - | - | - | 69.8 | 43.6 | 63.5 | 75.1 | 69.5 | 74.9 | 79.7 | 51.9 | 73.1 | - |
| Adapter Zhu et al. (2025) | Resnet101 | 74.9 | 54.3 | 65.1 | 62.8 | 57.9 | 58.6 | 79.9 | 51.9 | 73.2 | - | - | - | - | - | - | - |
| MIB Cermelli et al. (2020) | ViT | - | - | - | - | - | - | 72.6 | 23.5 | 60.9 | 80.4 | 47.8 | 78.8 | 78.5 | 63.2 | 74.9 | - |
| SSUL† Cha et al. (2021a) | ViT | 74.3 | 51.0 | 63.2 | 60.3 | 40.6 | 43.4 | 78.1 | 33.4 | 67.5 | 80.8 | 31.5 | 78.5 | 79.7 | 55.3 | 73.9 | 65.3 |
| MicroSeg† Zhang et al. (2022b) | ViT | 73.5 | 53.0 | 63.7 | 64.8 | 43.4 | 46.5 | 80.5 | 40.8 | 71.0 | 79.0 | 25.3 | 76.4 | 81.9 | 54.0 | 75.3 | 66.6 |
| CoinSeg Zhang et al. (2023) | ViT | 80.1 | 60.0 | 70.5 | 70.1 | 63.3 | 64.3 | 82.7 | 52.5 | 75.5 | 81.5 | 44.8 | 79.8 | 82.1 | 63.2 | 77.6 | 73.5 |
| MBS† Park et al. (2024) | ViT | 80.0 | 72.9 | 76.6 | 67.5 | 73.4 | 70.3 | 81.9 | 65.6 | 78.0 | 83.0 | 72.6 | 82.5 | 83.9 | 72.6 | 81.2 | 77.7 |
| Nest Xie et al. (2024) | ViT | 65.2 | 35.8 | 51.2 | - | - | - | 77.0 | 53.3 | 71.4 | 79.7 | 60.0 | 78.8 | 81.2 | 67.4 | 77.9 | - |
| Adapter-T Zhu et al. (2025) | ViT | - | - | - | - | - | - | 83.3 | 60.1 | 77.8 | - | - | - | - | - | - | - |
| Ours | ViT | 81.7 | 72.8 | 77.5 | 74.0 | 75.2 | 75.0 | 83.4 | 66.1 | 79.3 | 82.8 | 74.1 | 82.4 | 84.8 | 76.4 | 82.8 | 79.4 |

● Aeroplane  ● Bicycle  ● Bird  ● Boat  ● Bottle  ● Bus  ● Car  ● Cat  ● Chair  ● Cow
● Dining table  ● Dog  ● Horse  ● Motorbike  ● Person  ● Potted plant  ● Sheep  ● Sofa  ● Train  ● Tvmonitor

**Old classes from the base stage**

**New classes from the incremental stage**

Image    MIB    LGKD    Coinseg    MBS    Ours    GT

(a)

LGKD    Coinseg

MBS    Ours

(b)

Figure 4: (a) Visual comparison under the 15-1 setting. (b) T-SNE visualizations under the 15-1 setting. Our method achieves more accurate pixel-level segmentation of old classes with strong resistance to forgetting, while also reducing misclassification of new-class pixels. The t-SNE results further demonstrate that our method achieves more compact intra-class distributions and more dispersed inter-class distributions, indicating consistently low class-specific distribution overlap and effectively mitigating class confusion.

classes, misclassifying "sheep" as "horse" or "cow"—classes seen during the base step. Our method better preserves old knowledge and adapts more effectively to new classes.

**Qualitative analysis of class-specific knowledge distribution and confusion.** As shown in Fig. 4(b), we visualize the distributions of class-specific feature using t-SNE Van der Maaten & Hinton (2008) for our method and recent approaches under the 15-1 incremental setting. The plots reveal both intra-class compactness and inter-class separability. Compared to prior methods, our approach yields tighter clustering of features within the same class, reflected by denser colored point groups, and more distinct separation between different classes. This improved feature distribution reduces inter-class entanglement and effectively mitigates class confusion. For example, the "person" class (in pink) forms a noticeably more compact and isolated cluster, indicating enhanced class discrimination in incremental learning scenarios.

**Training time and convergence.** While our approach achieves consistent effectiveness across various incremental settings without increasing inference time, it introduces a slight training overhead for single-class learning: about 7 seconds per epoch longer than CKD on a single GPU (MKD: 51s, CKD: 53s, DKD: 60s). To evaluate convergence, we analyze the losses of MKD, CKD, and DKD in step 1 of the 19-1 incremental setting (Fig. 5). All methods show a rapid loss decline in the early epochs, but MKD and CKD exhibit higher initial loss and greater fluctuations, indicating less stable

Table 2: Comparative experiments on ADE20K Zhou et al. (2017). Our method is capable of effectively learning new knowledge and resisting catastrophic forgetting without accessing old-class data for rehearsal. Notably, the average performance of our method across the four incremental settings is very close to that of joint training, which is commonly regarded as the upper bound of performance in CISS.

| Method | Backbone | 100-5 (11 steps) | | | 100-10 (6 steps) | | | 50-50 (3 steps) | | | 100-50 (2 steps) | | | Average |
|---|---|---|---|---|---|---|---|---|---|---|---|---|---|---|
| | | Old | New | All | Old | New | All | Old | New | All | Old | New | All | |
| Joint | ViT | 49.5 | 38.0 | 45.7 | 49.7 | 38.4 | 46.0 | 55.0 | 41.1 | 59.7 | 48.9 | 38.2 | 45.4 | 49.2 |
| SDR Michieli & Zanuttigh (2021) | ResNet101 | 36.7 | 5.7 | 26.4 | 28.9 | 11.7 | 23.2 | 42.9 | 25.4 | 39.9 | 37.5 | 25.5 | 33.5 | 30.8 |
| PLOP Douillard et al. (2021a) | ResNet101 | 39.1 | 7.8 | 28.7 | 40.5 | 13.6 | 31.6 | 48.8 | 21.0 | 37.5 | 41.9 | 14.9 | 33.0 | 32.7 |
| RCIL Zhang et al. (2022a) | ResNet101 | 38.5 | 11.5 | 29.6 | 39.3 | 17.7 | 32.1 | 48.3 | 24.6 | 40.9 | 42.3 | 18.8 | 34.5 | 34.3 |
| SSUL Cha et al. (2021a) | ResNet101 | 39.9 | 17.4 | 32.4 | 40.2 | 18.8 | 33.1 | 48.4 | 20.2 | 36.5 | 41.3 | 18.0 | 33.6 | 33.9 |
| REMINDER Phan et al. (2022) | ResNet101 | 36.1 | 16.4 | 29.6 | 39.0 | 21.3 | 33.1 | 47.1 | 20.4 | 36.3 | 41.6 | 19.2 | 34.2 | 33.3 |
| Microseg Zhang et al. (2022b) | ResNet101 | 40.4 | 20.5 | 33.8 | 41.5 | 21.6 | 34.9 | 48.6 | 24.8 | 41.2 | 40.2 | 18.8 | 33.1 | 35.8 |
| EWF Xiao et al. (2023) | ResNet101 | 41.4 | 13.4 | 32.1 | 41.5 | 16.3 | 33.2 | - | - | - | 41.2 | 21.3 | 34.6 | - |
| IDEC Zhao et al. (2023) | ResNet101 | 39.2 | 14.6 | 31.0 | 42.3 | 17.6 | 34.1 | 47.4 | 26.0 | 42.0 | 42.0 | 18.2 | 34.1 | 35.3 |
| GSC Cong et al. (2023) | ResNet101 | - | - | - | 40.8 | 17.6 | 33.1 | 46.2 | 26.2 | 41.8 | 42.4 | 19.2 | 34.7 | - |
| LAG Yuan et al. (2024) | ResNet101 | 40.0 | 17.2 | 32.5 | 41.0 | 18.7 | 33.6 | 47.7 | 26.1 | 42.0 | 41.6 | 19.7 | 34.3 | 35.6 |
| CoMasTRe Gong et al. (2024) | ResNet101 | 40.8 | 15.8 | 32.5 | 42.3 | 18.4 | 34.4 | - | - | - | 45.7 | 26.0 | 39.2 | - |
| Adapter Zhu et al. (2025) | ResNet101 | 42.6 | 18.0 | 34.5 | 42.9 | 19.9 | 35.3 | 49.3 | 27.3 | 44.0 | 43.1 | 23.6 | 36.6 | 37.6 |
| MIB† Cermelli et al. (2020) | ViT | 40.2 | 26.6 | 35.7 | 43.0 | 30.8 | 39.0 | 52.2 | 35.6 | 53.2 | 46.4 | 35.0 | 42.6 | 42.6 |
| SSUL† Cha et al. (2021a) | ViT | 41.3 | 16.0 | 32.9 | 40.7 | 19.0 | 33.5 | 49.5 | 21.3 | 38.0 | 41.9 | 20.1 | 34.7 | 34.8 |
| Microseg† Zhang et al. (2022b) | ViT | 41.2 | 21.0 | 34.5 | 41.0 | 22.6 | 34.9 | 49.8 | 23.9 | 40.7 | 41.1 | 24.1 | 35.5 | 36.4 |
| Coinseg Zhang et al. (2023) | ViT | 43.1 | 24.1 | 36.8 | 42.1 | 24.5 | 36.3 | 49.0 | 28.9 | 45.4 | 41.6 | 26.7 | 36.7 | 38.8 |
| CoMFormer Cermelli et al. (2023) | ViT | 39.5 | 13.6 | 30.9 | 40.6 | 15.6 | 32.3 | - | - | - | 44.7 | 26.2 | 38.6 | - |
| INC Shang et al. (2023) | ViT | **46.9** | **31.3** | **41.7** | **48.5** | 34.6 | **43.9** | **56.2** | 37.8 | 56.8 | **49.4** | 35.6 | 44.8 | **46.8** |
| MBS† Park et al. (2024) | ViT | 45.7 | 22.7 | 38.1 | 48.1 | **35.2** | 43.8 | 55.6 | **39.8** | **58.6** | 49.4 | **37.6** | **45.5** | 46.5 |
| Ours | ViT | **47.2** | **30.0** | **41.5** | **48.7** | **37.3** | **44.9** | **56.6** | **40.5** | **59.6** | **49.3** | **39.9** | **46.2** | **48.1** |

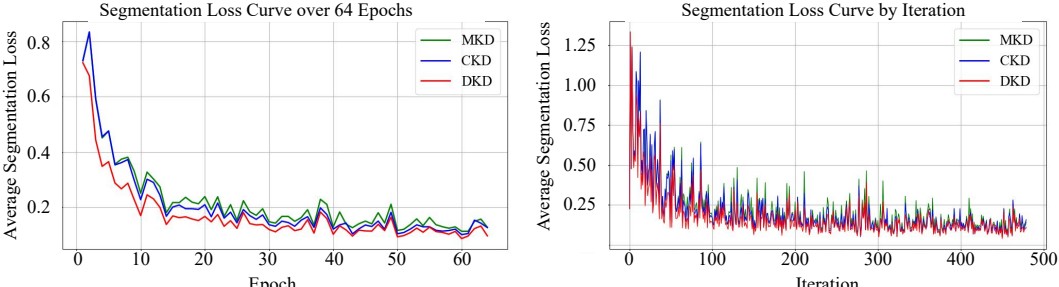

Figure 5: Epoch-wise(left) and iteration-wise loss curves (right). DKD stabilizes quickly and maintains a smoother trajectory, indicating better training stability and reliable convergence.

optimization. In contrast, DKD consistently achieves the lowest average loss and stabilizes quickly, demonstrating superior training stability and robustness in class-incremental semantic segmentation.

### 4.3 ABLATION STUDIES

**Impact of Adjusting the Distribution of Old Knowledge.** To evaluate the effectiveness of adaptive adjustment to the old model, we conduct an ablation study under the 10-1 setting. As shown in Fig. 6(a), the narrower orange shaded area compared to the blue one demonstrates that $\mathcal{L}_{\text{Min}}$ effectively balances the performance between old and new classes. In addition, the performance of the new classes improves significantly after incorporating $\mathcal{L}_{\text{Min}}$, as indicated by ✕. This suggests that dynamic adjustment of the old model's parameters facilitates the learning of new knowledge.

**Effectiveness of components in DKD.** To assess the contributions of DKD's loss components, we conduct an ablation study under the 10-1 setting, as shown in Tab. 3. Comparison of Grp. 1 with Grps. 2–4 reveals that each loss individually contributes to a slight improvement in the performance of both old and new classes. Comparisons between Grp. 2 and Grp. 5, as well as Grp. 2 and Grp. 6, show that combining $\mathcal{L}_{\text{Min}}$ with $\mathcal{L}_{\text{Esti}}$ or $\mathcal{L}_{\text{Max}}$ enhances resistance to catastrophic forgetting and improves plasticity. The comparison between Grp. 7 vs. Grp. 8 indicates that adjusting the old knowledge by $\mathcal{L}_{\text{Min}}$ allows better adaptation to new classes. To ensure the best overall performance across old and new classes, Grp. 8 is adopted as the configuration in this paper.

Table 3: Ablation study on the individual loss of DKD.

| Grp. | $\mathcal{L}_{\text{Min}}$ | $\mathcal{L}_{\text{Esti}}$ | $\mathcal{L}_{\text{Max}}$ | 10-1 | | |
|---|---|---|---|---|---|---|
| | | | | Old | New | All |
| 1 | ✗ | ✗ | ✗ | 63.5 | 21.3 | 43.4 |
| 2 | ✓ | | | 69.9 | 46.7 | 58.9 |
| 3 | | ✓ | | 72.7 | 45.2 | 59.6 |
| 4 | | | ✓ | 68.7 | 35.7 | 53.0 |
| 5 | ✓ | ✓ | | 82.1 | 70.8 | 76.7 |
| 6 | ✓ | | ✓ | 82.1 | 71.2 | 76.9 |
| 7 | | ✓ | ✓ | 71.2 | 49.6 | 60.9 |
| 8 | ✓ | ✓ | ✓ | 81.7 | 72.8 | 77.5 |

**Impact of DKD.** To evaluate the effectiveness of the proposed DKD in reducing class confusion, we compute the class similarity matrix for the 10 incremental classes (from class 11 to 20) in the

Table 4: Ablation study on the hyperparameter $\gamma$. In settings with fewer incremental classes, $\gamma$ indicates hyperparameter insensitivity; conversely, $\gamma = 0.4$ performs best.

| | 10-1 (11 steps) | | | | | | | 19-1 (2 steps) | | | | | | |
|---|---|---|---|---|---|---|---|---|---|---|---|---|---|---|
| | $\gamma=0$ | $\gamma=0.2$ | $\gamma=0.4$ | $\gamma=0.5$ | $\gamma=0.6$ | $\gamma=0.8$ | $\gamma=1.0$ | $\gamma=0$ | $\gamma=0.2$ | $\gamma=0.4$ | $\gamma=0.5$ | $\gamma=0.6$ | $\gamma=0.8$ | $\gamma=1.0$ |
| Old | 80.4 | 76.3 | 81.7 | **82.6** | 80.0 | 80.5 | 80.2 | 82.8 | 82.8 | 82.7 | **82.8** | **82.8** | 82.7 | **82.8** |
| New | 69.0 | 66.6 | **72.8** | 70.8 | 69.9 | 70.0 | 70.3 | 74.2 | **74.3** | 74.0 | **74.3** | 74.0 | 74.0 | 74.1 |
| All | 75.0 | 71.7 | **77.5** | 77.0 | 75.2 | 75.5 | 75.5 | **82.4** | 82.4 | 82.3 | **82.4** | **82.4** | 82.3 | **82.4** |

Figure 6: (a) compares the baseline with and without $L_{Min}$, showing that it enables adaptive adjustment of old parameters and knowledge distribution, facilitating new class learning while mitigating forgetting. (b) and (c) show that DKD lowers similarity between incrementally learned classes, reducing class confusion.

10-1 setting. This is shown in Fig. 6(b) (without DKD) and Fig. 6(c) (with DKD). In the figures, red indicates higher similarity between classes, while lighter red and blue represent lower similarity. As shown in Fig. 6(b), without DKD, the similarity between the classes learned during the incremental steps (classes 11–20) is relatively high, resulting in an overall similarity matrix with a reddish hue. However, after applying DKD, as shown in Fig. 6(c), the similarity between the learned classes significantly decreases. This suggests that the proposed DKD method effectively reduces class confusion, which in turn enhances pixel-level CISS performance.

**Ablation of hyperparameters.** Based on the experimental results from 14 sets of hyperparameters, as shown in Tab. 4, it is observed that for tasks with fewer incremental classes, such as 19-1, $\gamma$ has a minimal impact on the performance of both old and new classes, as well as their average performance. Thus, $\gamma$ in Eq. 6 is set to the default value of 1. As the number of incremental classes increases, new knowledge tends to be underfit without strengthened knowledge reuse; setting $\gamma$ too large or too small disrupts the performance balance between old and new knowledge. Thus, for tasks with more incremental classes, such as the 10-1, both the individual performance of old and new classes, as well as their average performance, reach optimal results when $\gamma$ is around 0.4. Accordingly, the default value of $\gamma$ is used for the 15-5, 15-1, and 19-1 tasks on Pascal VOC. For settings involving more incremental steps (10-1 and 2-2 on Pascal VOC) or a larger number of classes (100-50, 100-10, 50-50, and 100-5 on ADE20K), $\gamma$ is set to 0.4.

**Error Analysis.** As shown in Tab. 5, we evaluate experimental error by repeating the 10-1 configuration three times. The overall standard deviation for combined old and new class performance is approximately 0.1, confirming the stability and robustness of our metrics. Additional analysis is provided in Appendix C.7.

Table 5: Repeated experiments for error analysis.

| | 1 | 2 | 3 | Avg. | Std. |
|---|---|---|---|---|---|
| Old | 81.8 | 82.0 | 81.7 | 81.83 | 0.15 |
| New | 72.6 | 72.4 | 72.8 | 72.60 | 0.20 |
| All | 77.4 | 77.4 | 77.5 | 77.43 | **0.06** |

## 5 CONCLUSION

In this paper, we propose **D**istribution-based **K**nowledge **D**istillation (DKD), a minimization–maximization strategy designed to address parameter competition and knowledge reuse. DKD first releases low-sensitivity parameters of the old model and applies $\mathcal{L}_{Min}$ to minimize the old-knowledge distribution, thereby alleviating parameter competition under a static architecture. To better estimate the distribution of new knowledge and promote the reuse of acquired knowledge, we introduce $\mathcal{L}_{Esti}$ guided by Laplacian-based projection estimation. To further mitigate the underutilization of previously acquired knowledge, we then employ $\mathcal{L}_{Max}$ to maximize the shared knowledge distribution through an entropy-induced optimization. We establish the rationality of the method via theoretical analysis and demonstrate its effectiveness through extensive experiments. This minimization–maximization strategy reduces class confusion and achieves **near-upper-bound** average performance on ADE20K and Pascal VOC 2012, without incurring additional inference cost.

ETHICS STATEMENT

We adhere to the ICLR Code of Ethics in data use, experimentation, and manuscript preparation. The primary ethical considerations and compliance measures in this work are as follows.

**Human subjects and identifiable information.** This research does not involve human-subject experiments and does not collect or process any personally identifiable information (PII) or sensitive data. All experiments are conducted on publicly available datasets.

**Data and licensing.** We use the publicly available PASCAL VOC 2012 and ADE20K datasets under their original licenses and terms of use. Beyond standard preprocessing (e.g., label usage), we do not modify the data, synthesize new personal content, or inject information that could reveal identities.

**Potential harms.** Our continual-learning method can improve the stability and plasticity of semantic segmentation in long-term incremental scenarios, and could, in principle, be adapted to sensitive applications (e.g., surveillance). To reduce misuse risk: (1) we do not provide any data, models, or scripts tailored for face recognition, tracking, or other privacy-intrusive tasks; (2) we encourage practitioners to conduct application-level risk assessments (privacy, compliance, and security reviews) prior to deployment, and to use our method only in lawful, legitimate, and ethical contexts.

**Fairness, bias, and interpretability.** Our experiments follow standard CISS architectures and protocols from prior works, changing only the incremental-learning strategy. We provide theoretical analysis in the appendix to improve interpretability.

**Privacy and security.** We do not introduce raw images or metadata that could identify individuals. Training and evaluation do not involve inversion, re-identification, model stealing, or other high-risk procedures. Released code will not include scripts or interfaces for downloading personal data.

**Legal and regulatory compliance.** The research complies with dataset licenses and applicable copyright/usage terms. For any follow-up or regional extensions, users are responsible for ensuring compliance with local laws and ethical review requirements.

**Conflicts of interest and funding disclosure.** We have no undisclosed commercial conflicts of interest.

REPRODUCIBILITY STATEMENT

We are committed to ensuring the reproducibility of our work. In the following, we summarize the measures we have taken:

**Code Availability.** To facilitate the review process, the related code is included in the supplementary material. The released package includes model definitions, training scripts, and evaluation tools. Detailed instructions are also provided, covering the environment setup (Python version, PyTorch dependencies, and GPU drivers), execution commands, and hyperparameter settings, ensuring that the experiments can be easily replicated.

**Details of Experimental Setups.** The descriptions of the dataset, implementation details, metrics, baselines, and implementation configuration are provided in Appendix B. In particular, we specify the dataset splits, preprocessing steps, evaluation protocols, and the baseline implementations to guarantee fair comparisons. All hyperparameters used in training and testing are documented in both the main paper and the appendix.

**Computational Resources.** All experiments are conducted on six NVIDIA GeForce RTX 3090 GPUs and an Intel(R) Xeon(R) Gold 6226R CPU, with a batch size of 16, using PyTorch for implementation, as mentioned in Section 4.1 and Appendix B. We also record average training times and convergence to provide a clear view of the computational requirements in Section 4.2.

**Statistical Significance of Results.** To verify the robustness of our conclusions, we conduct multiple repeated experiments on complex tasks with a larger number of incremental steps. We observe that the average performance deviation across all classes remains around 0.1, as detailed in Section 4.3 and Appendix C.7. In addition, we provide variance and confidence interval analyses for the main results, which further demonstrate the stability and reliability of our method.

We believe these efforts ensure that our results can be reliably reproduced and extended by the research community.

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

APPENDIX

# A THEORETICAL ANALYSIS

## A.1 THEORETICAL ANALYSIS ABOUT $L_{Min}$

In this paper, to address the crowded distribution of old and new knowledge caused by parameter competition, we propose a strategy to minimize the discrepancy between the output of the parameter-released old model and the newly trained model over the old knowledge distribution in Section 3.2 (a). Here, the release is performed once per step for the old model, rather than at every epoch, and therefore does not introduce additional training overhead. Given the current training step $t$, the softmax output probability $y_c^t(h, w)$ represents the predicted probability of class $c$ at position $(h, w)$:

$$y_c^t(h, w) = \frac{\exp(z_c^t(h, w))}{\sum_j \exp(z_j^t(h, w))}, \tag{12}$$

the standard derivative of the softmax with respect to the unnormalized prediction score (logit) $z_k^t(h, w)$ for class $k$ at position $(h, w)$ is

$$\frac{\partial y_c^t}{\partial z_k^t} = y_c^t(\delta_{ck} - y_k^t), \tag{13}$$

where $\delta_{ck}$ is the Kronecker delta:

$$\delta_{ck} = \begin{cases} 1, & c = k, \\ 0, & c \neq k. \end{cases} \tag{14}$$

Thus, the loss gradient at the pixel $(h, w)$ of $L_{Min}$ can be written as:

$$\frac{\partial \ell(h, w)}{\partial z_k^t} = -\sum_{c \in C^{t-1}} y_c^* \frac{\partial y_c^t}{\partial z_k^t} \tag{15}$$

$$= -\sum_{c \in C^{t-1}} y_c^* y_c^t (\delta_{ck} - y_k^t), \tag{16}$$

**(i) When $k \in C^{t-1}$ (old classes):**

$$\frac{\partial \ell}{\partial z_k^t} = -y_k^* y_k^t (1 - y_k^t) + y_k^t \sum_{c \in C^{t-1}, c \neq k} y_c^* y_c^t, \tag{17}$$

where the first term encourages the current model to increase the probability $y_k^t$ for this old class, making it closer to the old model's output $y_k^*$, and the second term balances contributions from other old classes to maintain probability normalization. For old classes $k \in C^{t-1}$, the gradient direction encourages the current model's output at the retained parameter locations to not fall below the reference provided by the pruned old model, thereby achieving "soft target alignment".

**(ii) When $k \notin C^{t-1}$ (non-old classes):**

$$\frac{\partial \ell}{\partial z_k^t} = y_k^t \sum_{c \in C^{t-1}} y_c^* y_c^t, \tag{18}$$

this positive gradient indirectly shrinks the knowledge distribution for non-old classes, as the optimizer will adjust the logits during gradient descent to reduce this term, thereby alleviating crowded parameter-fitting knowledge space for old classes.

## A.2 THEORETICAL ANALYSIS ABOUT $L_{Esti}$

At pixel $(h, w)$, let $f := f_t(h, w) \in \mathbb{R}^D$ be the current feature and $p := \mathcal{P}_t(h, w) \in \mathbb{R}^D$ the "low-curvature coexistence point" inferred from the second-order information of $\mathcal{L}_{CKD}$. Let $y_c^*(h, w) \geq 0$ be the old-knowledge weight at this pixel. Define the pixel-wise distillation term $\ell_{CKD}(f) := y_c^*(h, w) \|f - f_{t-1}(h, w)\|_2$, the image-level distillation $\mathcal{L}_{CKD} := \frac{1}{HW} \sum_{h,w} \ell_{CKD}(f_t(h, w))$, and the Laplacian consistency $\mathcal{L}_{lap} := \frac{1}{HW} \sum_{h,w} \|f_t(h, w) - \mathcal{P}_t(h, w)\|_2$. Here $\|\cdot\|_2$ denotes the Euclidean norm. For any pixel,

$$\ell_{CKD}(f) - \ell_{CKD}(p) = y_c^*(h, w)\Big(\|f - f_{t-1}\|_2 - \|p - f_{t-1}\|_2\Big) \leq y_c^*(h, w) \|f - p\|_2, \tag{19}$$

where the inequality follows from the *reverse triangle inequality* $\|u\|_2 - \|v\|_2 \leq \|u - v\|_2$ with $u = f - f_{t-1}$ and $v = p - f_{t-1}$. Averaging over all pixels and letting $y_{\max} := \max_{h,w} y_c^*(h, w)$

yields

$$\mathcal{L}_{CKD}(f_t) - \mathcal{L}_{CKD}(\mathcal{P}_t) \ \le \ \frac{1}{HW}\sum_{h,w} y_c^*(h,w)\,\|f_t - \mathcal{P}_t\|_2 \ \le \ y_{\max}\,\mathcal{L}_{lap}. \tag{20}$$

Decreasing $\mathcal{L}_{lap}$ guarantees at least a *linear* reduction of the distillation gap, with proportionality governed by $y_c^*$—this is the direct, mathematical form of "reusing old knowledge."

Fix a pixel $(h, w)$ and write

$$f \equiv f_t(h,w) \in \mathbb{R}^D, \tag{21}$$

$$y \equiv y_c^*(h,w) \in \mathbb{R}^D, \tag{22}$$

$$\hat{f} := \frac{f}{\|f\|_2}. \tag{23}$$

$\mathcal{C}_t(h, w)$ preserves the degree of reusable old knowledge

$$\mathcal{C}_t(h,w) \ = \ \phi(f) \ := \ \frac{\langle y,\, f\rangle}{\|f\|_2}, \tag{24}$$

and denote a scalar weight by $\hat{w}$. The pixel-wise part of $L_{\text{pro}}$ is then

$$J(f) \ = \ -\hat{w}\,\mathcal{C}_t(h,w) \ = \ -\hat{w}\,\phi(f), \tag{25}$$

which matches the global definition;

$$L_{\text{pro}} = 1 - \frac{1}{HW}\sum_{h,w} \hat{w}(h,w)\,\mathcal{C}_t(h,w), \tag{26}$$

$$\mathcal{C}_t(h,w) = \frac{\langle y_c^*(h,w),\, f_t(h,w)\rangle}{\|f_t(h,w)\|_2}. \tag{27}$$

**(i) Gradient of $\mathcal{C}_t(h, w)$.** Write $\phi(f) = N/D$ with $N := y^\top f$ and $D := \|f\|_2 = (f^\top f)^{1/2}$. Then $\nabla_f N = y$ and $\nabla_f D = f/\|f\|_2 = \hat{f}$. By the quotient rule,

$$\nabla_f \phi(f) = \frac{D\,\nabla_f N - N\,\nabla_f D}{D^2} = \frac{\|f\|_2\, y - (y^\top f)\,\hat{f}}{\|f\|_2^2}$$

$$= \frac{1}{\|f\|_2}\Big(y - (\hat{f}^\top y)\,\hat{f}\Big) = \frac{1}{\|f\|_2}\,\big(I - \hat{f}\,\hat{f}^\top\big)\,y. \tag{28}$$

**(ii) Descent direction for the pixel-wise objective.** Since $J(f) = -\hat{w}\,\phi(f)$ with $\hat{w}$ treated as a local constant,

$$\nabla_f J(f) \ = \ -\hat{w}\,\nabla_f \phi(f) \ = \ -\frac{\hat{w}}{\|f\|_2}\,\big(I - \hat{f}\,\hat{f}^\top\big)\,y. \tag{29}$$

A gradient descent step $f^+ = f - \eta\,\nabla_f J(f)$ $(\eta > 0)$ yields the increment

$$\Delta f \ := \ f^+ - f \ = \ \eta\,\frac{w}{\|f\|_2}\,\big(I - \hat{f}\,\hat{f}^\top\big)\,y \ \implies \ \boxed{\ \Delta f \ \propto \ \big(I - \hat{f}\,\hat{f}^\top\big)\,y\ }. \tag{30}$$

This indicates that our update direction is the steepest ascent direction of the projection-based alignment, thereby aligning the current representation—without increasing its norm—under the guidance of the knowledge. The larger $\hat{w}(h,w)$ emphasizes high-confidence regions where old knowledge can be reused, guiding the update equation 30 to align the features with old knowledge. In contrast, low-confidence regions are scarcely affected, preserving capacity for new knowledge.

### A.3 THEORETICAL ANALYSIS ABOUT $L_{Max}$

Suppose $y_{(t,c)}^b \in [0, 1]$ denotes the predicted probability (softmax output) of sample $b \in \{1, \ldots, B\}$ at step $t$ for class $c \in \{1, \ldots, C_t\}$, with $\sum_{c=1}^{C_t} y_{(t,c)}^b = 1$. Define the batch marginal (per class)

$$\overline{y}_{(t,c)} = \frac{1}{B}\sum_{b=1}^{B} y_{(t,c)}^b, \tag{31}$$

the marginal entropy

$$H(Y_t) = -\sum_{c=1}^{C_t} \overline{y}_{(t,c)} \log \overline{y}_{(t,c)}, \tag{32}$$

and the conditional entropy proxy

$$H(Y_t \mid Y_{t-1}) = -\frac{1}{B} \sum_{b=1}^{B} \sum_{c=1}^{C_t} y_{(t,c)}^b \log y_{(t,c)}^b. \tag{33}$$

The objective is

$$\mathcal{L}_{\text{Max}} = -\frac{H(Y_t) - H(Y_t \mid Y_{t-1})}{\log |C_{t-1}|} = -\frac{I(Y_t; Y_{t-1})}{\log |C_{t-1}|}. \tag{34}$$

**(i) Gradient of the marginal entropy (chain rule).**
$$\frac{\partial H(Y_t)}{\partial y_{(t,c)}^b} = \frac{1}{B}\Big(-1 - \log \overline{y}_{(t,c)}\Big). \tag{35}$$

**(ii) Gradient of the conditional entropy (per-sample derivative).**
$$\frac{\partial H(Y_t \mid Y_{t-1})}{\partial y_{(t,c)}^b} = -\frac{1}{B}\Big(1 + \log y_{(t,c)}^b\Big). \tag{36}$$

**(iii) Gradient of $\mathcal{L}_{\text{Max}}$ (difference and normalization).** Combining equation 35–equation 36,

$$\boxed{\frac{\partial \mathcal{L}_{\text{Max}}}{\partial y_{(t,c)}^b} = -\frac{1}{B \log |C_{t-1}|} \log \frac{y_{(t,c)}^b}{\overline{y}_{(t,c)}}} \tag{37}$$

$$y_{(t,c)}^{b,\ \text{new}} \approx y_{(t,c)}^b + \eta \frac{1}{B \log |C_{t-1}|} \log \frac{y_{(t,c)}^b}{\overline{y}_{(t,c)}}. \tag{38}$$

If $y_{(t,c)}^b > \overline{y}_{(t,c)}$, then $\log \frac{y}{\overline{y}} > 0$ so $\partial \mathcal{L}_{\text{Max}}/\partial y < 0$; a gradient descent step $y \leftarrow y - \eta \, \partial \mathcal{L}_{\text{Max}}/\partial y$ ($\eta > 0$) *increases* $y$, making the per-sample distribution *sharper* (lower $H(Y_t \mid Y_{t-1})$). If $y_{(t,c)}^b < \overline{y}_{(t,c)}$, the sign reverses and $y$ decreases; aggregated over the batch, this pushes $\overline{y}$ toward *balance* (higher $H(Y_t)$), mitigating collapse. Since $\mathcal{L}_{\text{Max}} = -I(Y_t; Y_{t-1})/\log |C_{t-1}|$, minimizing $\mathcal{L}_{\text{Max}}$ is equivalent to maximizing $I(Y_t; Y_{t-1})$: it enforces *low conditional entropy* (sample-wise certainty given old knowledge) and *high marginal entropy* (batch-wise class balance). The former injects old discriminative knowledge into current predictions (retention/transfer), while the latter preserves capacity and diversity for new classes. Together, the update sharpens per-sample distributions and balances class usage across the batch—maximizing shared information between old and new distributions and thereby enabling "old-to-new" knowledge reuse.

# B  DETAILS OF EXPERIMENTAL SETUPS

**Dataset.** Following previous works Cermelli et al. (2020); Yang et al. (2023); Zhang et al. (2023); Park et al. (2024), our method is evaluated on the Pascal VOC 2012 Everingham et al. (2010) and ADE20K Zhou et al. (2017) datasets. The Pascal VOC dataset Everingham et al. (2010), a widely used benchmark for incremental segmentation, comprises 10,582 training images and 1,449 validation images across 20 categories. Additionally, the more challenging ADE20K dataset Zhou et al. (2017) includes 150 categories, with 20,210 training images and 2,000 validation images.

**Implementation details.** Following established practices Park et al. (2024); Zhang et al. (2023), we use ViT pretrained on ImageNet-1K as the backbone network. The decoder is composed of two transformer blocks, processing image patches at a fixed resolution of 512×512. Model optimization is conducted using stochastic gradient descent (SGD) with a momentum of 0.9 and a weight decay of $1 \times 10^{-5}$. Each training step in both the base and incremental steps is conducted for 64 epochs. The learning rate starts at $1 \times 10^{-3}$ in the base step ($t = 1$) and is adaptively adjusted in subsequent incremental steps ($t \geq 1$). For Pascal VOC, the learning rate is reduced to $1 \times 10^{-4}$ in the incremental steps, while for ADE20K, it is set to $5 \times 10^{-4}$. We set $\tau = 0.1$ for all experiments, and $\gamma$ is initialized to 1. For tasks with more incremental classes, including the 10-1 and 2-2 configuration of VOC and all incremental configurations of ADE20K, $\gamma$ is set to 0.4. All experiments are conducted on 6 Nvidia GeForce RTX 3090 GPUs and an Intel(R) Xeon(R) Gold 6226R CPU, with PyTorch used for implementation.

**Metrics.** In this paper, the performance of incremental semantic segmentation is evaluated using the commonly adopted mean Intersection over Union (mIoU) metric, consistent with previous methods. In the supplemental material, we additionally report Accuracy (ACC) to compare performance differences among subcategories. In our terminology, "Old" refers to the mIoU of old classes from the base step, "New" denotes the mIoU of new classes introduced in incremental steps, and "All" represents the mIoU across all classes, including background.

**Baselines.** In CISS, most current methods Cermelli et al. (2020); Yang et al. (2023); Zhang et al. (2023); Park et al. (2024) focus on optimizing incremental learning strategies rather than the architectural design. The primary architectural difference among existing methods lies in the choice of backbone—typically either ResNet101 or ViT. In this paper, we compare our method with both ResNet101-based and ViT-based approaches on the VOC and ADE20K datasets. To demonstrate that our method is not limited to ViT-based architectures, we replace the backbone in CoinSeg with ResNet101 and apply our proposed DKD strategy for performance validation. We also report re-implementation results of existing methods with ViT as the backbone, such as MBS† and SSUL†. Additionally, we provide the performance of our architecture under joint training across multiple incremental configurations, which is currently considered the upper bound of incremental learning for models in this field Yang et al. (2023); Baek et al. (2022).

**Incremental configurations.** In addition to the widely explored incremental configurations on the VOC dataset—15-1, 15-5, and 19-1. This paper also conducts experiments on configurations with more incremental steps, such as 10-1 and 2-2. Furthermore, we evaluate our method on four configurations on ADE20K: 100-50, 50-50, 100-10, and 100-5. These diverse experimental setups allow for a comprehensive assessment of the robustness of our approach under various incremental learning settings. For example, the 19-1 setting represents a typical setting where most foreground classes (19 in total, excluding background) are learned in the base step, with only one new class introduced in the incremental step. In the 2-2 configuration, the model is initially trained on 2 foreground classes, with 2 additional classes added at each of the 9 subsequent incremental steps—culminating in 18 new classes. This setup reflects a representative case of a substantial number of classes to be acquired incrementally.

# C  Additional Experiments

## C.1  Performance comparison in the Disjoint setting

As shown in Tab. 6, in addition to the commonly explored overlap setting, we also conduct a quantitative analysis on the VOC dataset using an additional disjoint setting. In the overlap setting, the background includes previously learned classes, the classes to be learned at the current step, and the classes to be learned in future steps. In the disjoint setting, the background excludes classes required for future steps. Taking into account both old and new classes, our method achieves the best results in all three incremental configurations under the disjoint setting, with overall improvements of 0.2%, 0.1%, and 0.9%, respectively. In cases with fewer incremental steps and fewer new classes to learn, our method demonstrates better learning of new classes. Specifically, in the 19-1 task, our method improves the MIoU metric by 5.2%. In the 15-1 task, with more incremental steps, our method achieves a 4% improvement on new classes. This demonstrates that our method is not only effective for the overlap setting mentioned in Tab. 1 in the main paper, but also excels in CISS under the disjoint setting, showing both resistance to forgetting old classes and adaptability to new ones.

## C.2  Performance comparison on individual classes

To analyze the impact of our method on the 20 foreground classes in VOC under the widely explored overlap setting, we compute the MIoU values of the 20 subcategories for both our method and the MBS method across three different incremental configurations. Considering the average performance across the 20 foreground classes as shown in Tab. 7, our method achieves the best results in multiple incremental configurations, with improvements of 1.6%, 1.3%, and 2.5%, respectively. In the 2-2 task with more incremental steps, our method demonstrates significant resistance to forgetting, particularly for the "Aeroplane" and "Bottle" classes, with improvements of 17.8% and 29.0%, respectively. Additionally, in the 15-5 incremental configuration, our method improves the performance of new classes—"Potted plant", "Sheep", "Sofa", "Train", and "TV monitor"—by 3.2%, 2.1%, 5.6%, 4.0%,

Table 6: Performance comparison of the disjoint setting on the VOC dataset. Unlike the overlapping setting, which has been widely explored in Tab. 1 of the main paper, the disjoint setting only includes the classes from the current step $C_t$ and the previous old classes $C_{1:t-1}$, without including any future classes. The optimal and suboptimal performances are respectively represented in **red** and **blue** bold. The † symbol indicates results reproduced following the publicly released code. Our method demonstrates notable performance gains across most incremental configurations, including a $5.2\%$ improvement in new classes for 19-1 and a $4\%$ boost in new classes for 15-1.

| Method | Publication | Backbone | 19-1 (2 steps) Old | New | All | 15-5 (2 steps) Old | New | All | 15-1 (6 steps) Old | New | All |
|---|---|---|---|---|---|---|---|---|---|---|---|
| Joint | - | ViT | 84.5 | 80.6 | 84.3 | 85.3 | 80.7 | 84.2 | 85.3 | 80.4 | 84.1 |
| MiB Cermelli et al. (2020) | CVPR 2020 | ResNet101 | 69.6 | 25.6 | 67.5 | 71.8 | 43.3 | 65.0 | 46.2 | 12.9 | 38.3 |
| MiB† Cermelli et al. (2020) | CVPR 2020 | ViT | **80.6** | 45.2 | 78.9 | 75.0 | 59.9 | 71.4 | 66.7 | 26.3 | 57.1 |
| SDR Michieli & Zanuttigh (2021) | CVPR 2021 | ResNet101 | 69.9 | 37.3 | 68.3 | 73.5 | 47.3 | 67.3 | 59.2 | 12.9 | 48.2 |
| PLOP Douillard et al. (2021a) | CVPR 2021 | ResNet101 | 75.4 | 38.9 | 73.7 | 71.0 | 42.8 | 64.3 | 57.9 | 13.7 | 47.4 |
| RBC Zhao et al. (2022) | ECCV 2022 | ResNet101 | 76.4 | 45.8 | 74.9 | 75.1 | 49.7 | 69.1 | 61.7 | 19.5 | 51.7 |
| RBC† Zhao et al. (2022) | ECCV 2022 | ViT | 80.9 | 42.1 | 79.1 | 77.7 | 59.1 | 73.3 | 69.0 | 28.4 | 59.3 |
| MBS† Park et al. (2024) | ECCV 2024 | ViT | 84.4 | **70.8** | **83.8** | **82.7** | **68.6** | **79.3** | **81.0** | **62.0** | **76.5** |
| Ours | - | ViT | 84.4 | **76.0** | **84.0** | **82.8** | **68.5** | **79.4** | **80.9** | **66.0** | **77.4** |

and $4.3\%$, respectively. Based on the above analysis, our method is shown to be effective not only for configurations with fewer incremental steps (e.g., 15-5) but also for tasks with more incremental steps (e.g., 2-2).

Table 7: Performance comparison of the overlapping setting between our method and the recent MBS method on 20 individual classes in VOC (excluding the background class). ↑ and ↓ represent the magnitude of improvement and decline in MIoU, respectively, when compared to MBS Park et al. (2024). Considering the 20 foreground classes in VOC, our method achieves the best multi-class average performance in all 3 configurations, with improvements of $1.6\%$, $1.3\%$, and $2.5\%$, respectively.

| Class Name | MBS Park et al. (2024) 15-5 | 15-1 | 2-2 | Ours 15-5 | 15-1 | 2-2 | Comparison 15-5 | 15-1 | 2-2 |
|---|---|---|---|---|---|---|---|---|---|
| Aeroplane | 94.3 | 93.7 | 68.3 | 94.9 | 92.4 | 86.1 | ↑0.6 | ↓1.3 | ↑17.8 |
| Bicycle | 45.5 | 46.3 | 42.6 | 44.6 | 44.6 | 43.8 | ↓0.9 | ↓1.7 | ↑1.2 |
| Bird | 92.1 | 83.0 | 80.4 | 94.1 | 86.5 | 87.1 | ↑2.0 | ↑3.5 | ↑6.7 |
| Boat | 80.9 | 77.3 | 60.6 | 81.3 | 80.5 | 59.0 | ↑0.4 | ↑3.2 | ↓1.6 |
| Bottle | 84.1 | 85.6 | 54.6 | 88.5 | 85.4 | 83.6 | ↑4.4 | ↓0.2 | ↑29.0 |
| Bus | 92.3 | 92.8 | 80.5 | 93.5 | 95.2 | 81.7 | ↑1.2 | ↑2.4 | ↑1.2 |
| Car | 92.1 | 89.4 | 82.1 | 92.2 | 91.6 | 80.1 | ↑0.1 | ↑2.2 | ↓2.0 |
| Cat | 95.3 | 93.7 | 94.9 | 95.5 | 94.7 | 95.1 | ↑0.2 | ↑1.0 | ↑0.2 |
| Chair | 52.7 | 46.1 | 44.9 | 51.6 | 53.7 | 45.5 | ↓1.1 | ↑7.6 | ↑0.6 |
| Cow | 95.2 | 95.7 | 91.7 | 95.2 | 94.0 | 89.3 | 0.0 | ↓1.7 | ↓2.4 |
| Dining table | 61.9 | 55.2 | 59.1 | 63.9 | 60.0 | 60.1 | ↑2.0 | ↑4.8 | ↑1.0 |
| Dog | 91.6 | 88.7 | 88.8 | 93.5 | 91.9 | 91.6 | ↑1.9 | ↑3.2 | ↑2.8 |
| Horse | 93.3 | 92.5 | 89.0 | 93.3 | 91.8 | 88.0 | 0.0 | ↓0.7 | ↓1.0 |
| Motorbike | 89.4 | 89.7 | 86.7 | 90.0 | 90.4 | 83.4 | ↑0.6 | ↑0.7 | ↓3.3 |
| Person | 89.4 | 88.6 | 89.6 | 90.2 | 89.7 | 90.2 | ↑0.8 | ↑1.1 | ↑0.6 |
| Potted plant | 68.4 | 60.2 | 67.8 | 71.6 | 63.2 | 65.8 | ↑3.2 | ↑3.0 | ↓2.0 |
| Sheep | 89.9 | 86.5 | 78.4 | 92.0 | 85.2 | 76.0 | ↑2.1 | ↓1.3 | ↓2.4 |
| Sofa | 45.4 | 35.1 | 36.8 | 51.0 | 38.8 | 39.0 | ↑5.6 | ↑3.7 | ↑2.2 |
| Train | 86.4 | 81.3 | 64.2 | 90.4 | 80.8 | 67.6 | ↑4.0 | ↓0.5 | ↑3.4 |
| Tv monitor | 73.0 | 64.7 | 72.7 | 77.3 | 62.3 | 70.5 | ↑4.3 | ↓2.4 | ↓2.2 |
| Avarage | 80.7 | 77.3 | 71.7 | 82.2 | 78.6 | 74.2 | ↑1.6 | ↑1.3 | ↑2.5 |

### C.3  ABLATION STUDY OF THE BACKBONE

To further assess the generalizability and compatibility of our proposed DKD strategy beyond ViT-based architectures, we also evaluate its effectiveness when integrated into a ResNet101-based CISS framework. In particular, we adopt CoinSeg, built upon ResNet101 as backbone, and incorporate our DKD module. We then conduct experiments under the 19-1 task setting to measure its performance. As shown in Tab. 8, we report the detailed performance of the original CoinSeg (with ResNet101 backbone) and CoinSeg enhanced with DKD across all 20 foreground classes and the background. For the new class "Tv monitor", the original CoinSeg achieves a MIoU of $43.5\%$. After integrating DKD, the performance improves significantly to $48.3\%$, resulting in a $4.8\%$ gain. Averaged over all 20 foreground classes and background, the ResNet101-based architecture obtains a $0.3\%$ increase in MIoU and a $0.5\%$ improvement in overall accuracy (ACC) with DKD. These results clearly demonstrate that our DKD strategy is not limited to ViT-based models, but also exhibits compatibility and orthogonality with ResNet101-based architectures.

Table 8: Performance Comparison on CoinSeg (ResNet101) and CoinSeg + DKD (ResNet101). When applied to architectures with ResNet101 as the backbone, our method yields substantial gains in the segmentation performance of newly introduced categories. Specifically, under the 19-1 setting, the mIoU for the new class (TV monitor) improves by $4.8\%$, highlighting the adaptability and effectiveness of our approach beyond ViT-based models.

| Class Name | Coinseg (ResNet101) | | Coinseg + DKD (ResNet101) | |
|---|---|---|---|---|
| | MIoU | Acc | MIoU | Acc |
| Background | 92.0 | 95.6 | 92.2 | 95.7 |
| Aeroplane | 90.2 | 95.8 | 89.1 | 96.1 |
| Bicycle | 38.3 | 89.2 | 37.1 | 90.5 |
| Bird | 91.2 | 95.9 | 91.5 | 96.3 |
| Boat | 68.6 | 88.5 | 68.0 | 90.2 |
| Bottle | 76.3 | 92.8 | 76.3 | 93.2 |
| Bus | 92.3 | 95.6 | 93.2 | 96.3 |
| Car | 86.2 | 92.8 | 85.0 | 93.1 |
| Cat | 95.1 | 98.6 | 93.5 | 98.0 |
| Chair | 31.5 | 40.8 | 33.2 | 43.1 |
| Cow | 84.1 | 91.2 | 83.6 | 90.7 |
| Dining table | 55.1 | 62.4 | 56.2 | 65.4 |
| Dog | 88.5 | 95.0 | 87.6 | 94.1 |
| Horse | 79.9 | 88.8 | 80.1 | 90.4 |
| Motorbike | 87.5 | 95.5 | 87.7 | 95.9 |
| Person | 88.0 | 91.1 | 87.6 | 91.5 |
| Potted plant | 53.4 | 68.9 | 55.3 | 72.7 |
| Sheep | 76.1 | 89.9 | 76.4 | 89.6 |
| Sofa | 45.8 | 53.8 | 48.0 | 57.0 |
| Train | 88.4 | 92.7 | 88.9 | 93.3 |
| Tv monitor | 43.5 | 85.8 | 48.3 | 77.0 |
| Average | 73.9 | 85.7 | 74.2 | 86.2 |

### C.4  MORE EXPERIMENTS ABOUT THE PROPOSED DKD.

To evaluate the effectiveness of our method within other CISS methods, we integrate the proposed DKD strategy into the original CoinSeg, employing a ViT backbone under the 19-1 incremental setting. We then compare the performance with the original CoinSeg method. As shown in Tab. 9, by augmenting CoinSeg with our DKD while retaining its original learning strategy, the MIoU for the new class "Tv monitor" is significantly improved by $20.2\%$. Moreover, considering the average performance over all 20 foreground classes and the background class, our method brings an overall mIoU improvement of $1.3\%$ and an accuracy (ACC) gain of $0.6\%$. These results demonstrate that our DKD strategy not only enhances the model's ability to acquire new class knowledge but also helps preserve knowledge of previously learned classes by mitigating catastrophic forgetting. Besides, we also conducted related experiments on incremental remote sensing data. On the iSAID dataset, joint

training achieved upper bounds of 45.2 and 52.3 for old and new classes, respectively. Our method achieved 44.6 and 49.3, showing that it can perform close to joint training, which is considered the upper bound for incremental tasks.

Table 9: Performance Comparison on CoinSeg (ViT) and CoinSeg + DKD (ViT). The integration of the proposed DKD loss into the existing CoinSeg framework highlights its flexibility as a plug-and-play module. Notably, DKD boosts the mIoU of the novel class "TV monitor" by 20.2%.

| Class Name | Coinseg (ViT) | | Coinseg + DKD (ViT) | |
|---|---|---|---|---|
| | MIoU | Acc | MIoU | Acc |
| Background | 93.0 | 96.3 | 94.0 | 97.2 |
| Aeroplane | 88.1 | 92.1 | 90.5 | 95.0 |
| Bicycle | 42.9 | 86.6 | 42.7 | 89.7 |
| Bird | 95.1 | 97.1 | 95.1 | 97.4 |
| Boat | 74.9 | 88.0 | 75.6 | 89.1 |
| Bottle | 85.3 | 94.6 | 85.1 | 94.8 |
| Bus | 95.0 | 97.3 | 95.6 | 97.5 |
| Car | 90.5 | 92.4 | 90.3 | 92.4 |
| Cat | 96.2 | 98.3 | 96.2 | 98.3 |
| Chair | 48.1 | 59.5 | 49.0 | 62.3 |
| Cow | 93.8 | 97.7 | 93.7 | 98.0 |
| Dining table | 58.4 | 60.1 | 59.1 | 61.0 |
| Dog | 92.9 | 97.7 | 92.6 | 97.8 |
| Horse | 91.7 | 94.6 | 92.2 | 95.2 |
| Motorbike | 92.4 | 95.9 | 92.5 | 96.1 |
| Person | 89.7 | 92.6 | 89.8 | 93.0 |
| Potted plant | 68.6 | 85.0 | 68.6 | 85.8 |
| Sheep | 93.2 | 95.3 | 92.5 | 95.6 |
| Sofa | 59.5 | 67.9 | 60.1 | 68.6 |
| Train | 90.8 | 92.2 | 90.9 | 92.4 |
| Tv monitor | 38.6 | 86.1 | 58.8 | 82.8 |
| Average | 79.9 | 88.9 | 81.2 | 89.5 |

## C.5 ABLATION STUDY OF THRESHOLD $\tau$

Through the ablation study, we validate the necessity of releasing old parameters based on the value of $\tau$. Specifically, by comparing Groups (Grps.) 1 and 2 in Tab. 3 of the main paper, we observe that releasing old knowledge parameters with the assistance of $\tau$ and minimizing the old distribution can enhance the plasticity of new knowledge. To assess the reasonableness and impact of selecting the threshold $\tau$ during the release of old model parameters, we conduct experiments with five different values of $\tau$ on the 10-1 and 19-1 incremental configurations. As shown in Tab. 10, we evaluate the performance of different $\tau$ values on old classes (including the background class), new classes emerging in the incremental steps, and overall performance across all classes (including the background class). In the 10-1 task, when $\tau$ increases from 0.05 to 0.1, we observe a 0.3% performance improvement on new classes, while in the 19-1 task, the improvement reaches 1.6%. Thus, increasing $\tau$ can enhance the ability to learn new classes. By observing the performance changes of $\tau$ values between 0.15 and 0.4, we find that excessively large values of $\tau$ lead to the forgetting of old classes. Based on the experiments in Tab. 10, we conclude that if the priority is to preserve the performance on old classes, $\tau = 0.05$ is a good choice. However, if the focus is on learning new knowledge, $\tau = 0.1$ is a better option. For all experiments in this paper, we use $\tau = 0.1$, which is more favorable for learning new classes in the incremental steps.

To further validate that pruning with a predefined threshold does not lead to significant forgetting of previously learned classes, we conduct experiments under the 19-1 setting. Specifically, we first load the model at step 0 of the 19-1 task and evaluate its performance without pruning (W/O pruning), measuring the mIoU and ACC on the 19 classes already learned. Next, we apply pruning and re-evaluate the same model (W pruning) on the identical set of 19 classes. As shown in Tab. 11, the results indicate that this pruning strategy induces only negligible performance degradation.

Table 10: Ablation study on the threshold $\tau$. Considering both tasks with more incremental steps (e.g., 10-1) and tasks with fewer incremental steps (e.g., 19-1), we evaluate the resistance to forgetting for old classes, the ability to learn new classes, and the average performance across all classes (including the background class). Based on the performance on new classes, we select $\tau=0.1$ as the value for all experiments.

| | 10-1 | | | | | | 19-1 | | | | | |
| | $\tau = 0.05$ | $\tau = 0.1$ | $\tau = 0.15$ | $\tau = 0.2$ | $\tau = 0.3$ | $\tau = 0.4$ | $\tau = 0.05$ | $\tau = 0.1$ | $\tau = 0.15$ | $\tau = 0.2$ | $\tau = 0.3$ | $\tau = 0.4$ |
|---|---|---|---|---|---|---|---|---|---|---|---|---|
| Old | 83.1 | 81.7 | 81.8 | 81.6 | 81.1 | 76.9 | 83.1 | 82.8 | 82.6 | 82.4 | 82.4 | 1.1 |
| New | 72.5 | 72.8 | 72.9 | 72.1 | 64.5 | 49.4 | 72.5 | 74.1 | 72.8 | 72.3 | 71.9 | 3.1 |
| All | 78.1 | 77.5 | 77.6 | 77.1 | 73.2 | 63.8 | 82.6 | 82.4 | 82.1 | 81.9 | 81.9 | 1.2 |

Table 11: Impact of parameter pruning on the performance of learned classes. The results show that pruning leads to only a minor performance drop on the learned classes, indicating that it does not cause significant forgetting.

| Class Name | W pruning | | W/O pruning | |
|---|---|---|---|---|
| | MIoU | Acc | MIoU | Acc |
| Aeroplane | 93.2 | 97.5 | 94.3 | 97.6 |
| Bicycle | 43.9 | 89.6 | 45.2 | 90.5 |
| Bird | 93.1 | 96.2 | 94.8 | 97.6 |
| Boat | 78.5 | 91.0 | 78.2 | 91.8 |
| Bottle | 86.4 | 95.4 | 87.7 | 96.2 |
| Bus | 92.8 | 95.5 | 92.7 | 95.5 |
| Car | 88.8 | 95.7 | 89.8 | 96.3 |
| Cat | 94.4 | 97.1 | 95.4 | 97.4 |
| Chair | 51.7 | 67.4 | 55.7 | 74.6 |
| Cow | 95.0 | 97.2 | 95.3 | 97.7 |
| Dining table | 62.6 | 65.1 | 63.9 | 66.5 |
| Dog | 93.2 | 97.1 | 94.3 | 97.9 |
| Horse | 92.7 | 95.2 | 92.6 | 94.9 |
| Motorbike | 85.9 | 91.5 | 86.7 | 91.7 |
| Person | 88.5 | 93.6 | 89.3 | 94.4 |
| Potted plant | 71.7 | 85.8 | 72.5 | 96.4 |
| Sheep | 91.4 | 93.9 | 94.4 | 96.6 |
| Sofa | 57.2 | 63.2 | 57.6 | 63.2 |
| Train | 90.8 | 98.3 | 90.6 | 98.3 |
| Tv monitor | Unseen classes | Unseen classes | Unseen classes | Unseen classes |
| Average | 82.3 | 90.2 | 83.2 | 91.1 |

Furthermore, ablation results confirm that parameter release, under the guidance of $\mathcal{L}_{\text{Min}}$, effectively promotes the learning of new classes in subsequent steps (refer to Tab.12 and Sec. C.6 for more details). These findings demonstrate the effectiveness of our parameter release strategy.

## C.6 FURTHER EVALUATION OF DKD COMPONENT EFFECTIVENESS.

To evaluate the individual contributions of the 3 loss components ($\mathcal{L}_{\text{Min}}$, $\mathcal{L}_{\text{Esti}}$, and $\mathcal{L}_{\text{Max}}$) in DKD, we conduct an ablation study focusing on their respective roles. As shown in Tab. 12, experiments are conducted under 10-1, 15-1, and 19-1 settings to evaluate the effectiveness of the component in different complex incremental scenarios. Comparing Grp. 1 with Grps. 2–4 show that each loss function individually helps slightly mitigate the performance imbalance between old and new classes. After adding the $\mathcal{L}_{\text{Min}}$ loss, the new classes in the 10-1 setting, which involve more incremental steps, show a 25.4% improvement over the baseline Grp. 1. In settings with fewer incremental steps, such as 15-1 and 19-1, the performance on new classes improves by 4.1% and 3.3%, respectively. These results demonstrate that the adaptive adjustment and optimization of the old model's parameters during the distillation process significantly enhance the plasticity of new knowledge in continuous learning. Comparisons between Grp. 5 and Grp. 8, as well as Grp. 6 and Grp. 8, show that combining $\mathcal{L}_{\text{Min}}$ with either $\mathcal{L}_{\text{Esti}}$ or $\mathcal{L}_{\text{Max}}$ significantly improves the balance between old and new

Table 12: Ablation study of the components of the proposed DKD across multiple incremental configurations. Through various incremental learning setups, it is observed that each loss function helps promote a better balance between the performance on old and new classes.

| Grp. | $\mathcal{L}_{\text{Min}}$ | $\mathcal{L}_{\text{Esti}}$ | $\mathcal{L}_{\text{Max}}$ | 10-1 (11 Steps) | | | 15-1 (6 Steps) | | | 19-1 (2 Steps) | | | Average | | |
|---|---|---|---|---|---|---|---|---|---|---|---|---|---|---|---|
| | | | | Old | New | All | Old | New | All | Old | New | All | Old | New | All |
| 1 | ✗ | ✗ | ✗ | 63.5 | 21.3 | 43.4 | 74.9 | 38.3 | 66.2 | 83.3 | 66.7 | 82.5 | 73.9 | 42.1 | 64.0 |
| 2 | ✓ | | | 69.9 | 46.7 | 58.9 | 81.8 | 42.4 | 72.4 | 82.5 | 70.0 | 81.9 | 78.1 | 53.0 | 71.1 |
| 3 | | ✓ | | 72.7 | 45.2 | 59.6 | 81.0 | 36.6 | 70.4 | 82.9 | 69.7 | 82.3 | 78.9 | 50.5 | 70.8 |
| 4 | | | ✓ | 68.7 | 35.7 | 53.0 | 82.2 | 50.1 | 74.6 | 82.9 | 71.0 | 82.3 | 77.9 | 52.3 | 70.0 |
| 5 | ✓ | ✓ | | 82.1 | 70.8 | 76.7 | 82.2 | 68.0 | 78.8 | 82.6 | 69.8 | 82.0 | 82.3 | 69.5 | 79.2 |
| 6 | ✓ | | ✓ | 82.1 | 71.2 | 76.9 | 83.3 | 67.0 | 79.4 | 82.6 | 72.3 | 82.1 | 82.7 | 70.2 | 79.5 |
| 7 | | ✓ | ✓ | 71.2 | 49.6 | 60.9 | 81.2 | 39.9 | 71.4 | 82.9 | 70.2 | 82.3 | 78.4 | 53.2 | 71.5 |
| 8 | ✓ | ✓ | ✓ | 81.7 | 72.8 | 77.5 | 82.6 | 70.0 | 79.6 | 82.8 | 74.1 | 82.4 | 82.4 | 72.3 | 79.8 |

class performance, enhancing resistance to catastrophic forgetting and knowledge plasticity. The Grp. 7 vs. Grp. 8 comparison indicates that refining the old knowledge by $\mathcal{L}_{\text{Min}}$ allows better adaptation to new classes. Ablation experiments conducted under 3 different incremental settings effectively validate the role of the 3 distinct losses in class-incremental semantic segmentation. This further demonstrates that the combination of these three losses enhances the model's ability to learn new classes while resisting catastrophic forgetting of old classes across various incremental settings.

## C.7 FURTHER EXPERIMENTS OF ERROR ANALYSIS

Table 13: Error analysis. The overall performance error of the mean is approximately 0.1

| | 10-1 (11 steps) | | | | | 2-2 (10 steps) | | | | | 15-1(6 steps) | | | | | 19-1 (2 steps) | | | | | 15-5 (2 steps) | | | | |
|---|---|---|---|---|---|---|---|---|---|---|---|---|---|---|---|---|---|---|---|---|---|---|---|---|---|
| | 1 | 2 | 3 | Average | Std | 1 | 2 | 3 | Average | Std | 1 | 2 | 3 | Average | Std | 1 | 2 | 3 | Average | Std | 1 | 2 | 3 | Average | Std |
| Old | 81.8 | 82.0 | 81.7 | 81.83 | 0.15 | 74.0 | 74.1 | 74.0 | 74.03 | 0.06 | 83.4 | 83.2 | 83.3 | 83.30 | 0.10 | 82.8 | 82.8 | 82.6 | 82.73 | 0.12 | 84.8 | 84.8 | 84.7 | 84.77 | 0.06 |
| New | 72.6 | 72.4 | 72.8 | 72.60 | 0.20 | 75.2 | 74.7 | 75.0 | 74.97 | 0.25 | 66.1 | 66.4 | 66.0 | 66.17 | 0.21 | 74.0 | 74.1 | 74.0 | 74.03 | 0.06 | 76.4 | 76.2 | 76.2 | 76.27 | 0.12 |
| All | 77.4 | 77.4 | 77.5 | 77.43 | 0.06 | 75.0 | 74.6 | 74.9 | 74.83 | 0.21 | 79.3 | 79.2 | 79.2 | 79.23 | 0.06 | 82.4 | 82.4 | 82.2 | 82.33 | 0.12 | 82.8 | 82.8 | 82.7 | 82.77 | 0.06 |

As shown in Tab. 13, we evaluate the experimental variability by conducting repeated experiments under the 10-1 and 19-1 settings. Among them, 10-1 represents a complex incremental setting with numerous incremental steps, while 19-1 involves a greater number of classes learned during the base step. We select these two representative and challenging settings for error analysis. Each setting is run three times, recording the mIoU for old classes (Old), newly introduced classes (New), and all classes excluding the background (All). We report the mean and standard deviation (Std) across three repeated experiments. The results show that the overall variability for the combined performance of both old and new classes is consistently close to 0.1, further validating the stability and reliability of the performance under varying incremental learning conditions.

# D MORE QUALITATIVE ANALYSIS

To further validate the effectiveness of our proposed method in improving pixel-level semantic classification accuracy, we present more qualitative comparisons with recent state-of-the-art approaches in Figs. 7-10 under 15-1 setting. As illustrated in the second row of Fig. 7, existing methods tend to confuse the background "cloth" with the "sofa" class after learning sofa-related knowledge in the incremental step. In the third row, "windows" are mistakenly segmented as "TV monitor" or "sofa", indicating that these methods struggle to distinguish fine-grained class segmentation when integrating new knowledge. Our method maintains the stability of previously learned classes while effectively acquiring new concepts, demonstrating improved plasticity in adapting to new classes. Fig. 8 provides additional results, where the first row shows a case of "motorcycle" being largely forgotten after five incremental steps. Some methods fails to retain the object's contour and misclassifies it as other irrelevant categories. A similar confusion is more evident in Fig. 9 in the background of the first row. Our approach successfully preserves fine-grained class distinctions, significantly reducing pixel-level misclassification. In the final row of Fig. 10, existing methods confuse the "chair" occupied by a baby with a "sofa". Our method achieves much more precise segmentation, with only a small portion misclassified. These qualitative results clearly demonstrate that our approach substantially enhances pixel-wise classification accuracy in class-incremental semantic segmentation, while effectively mitigating class confusion during continual learning.

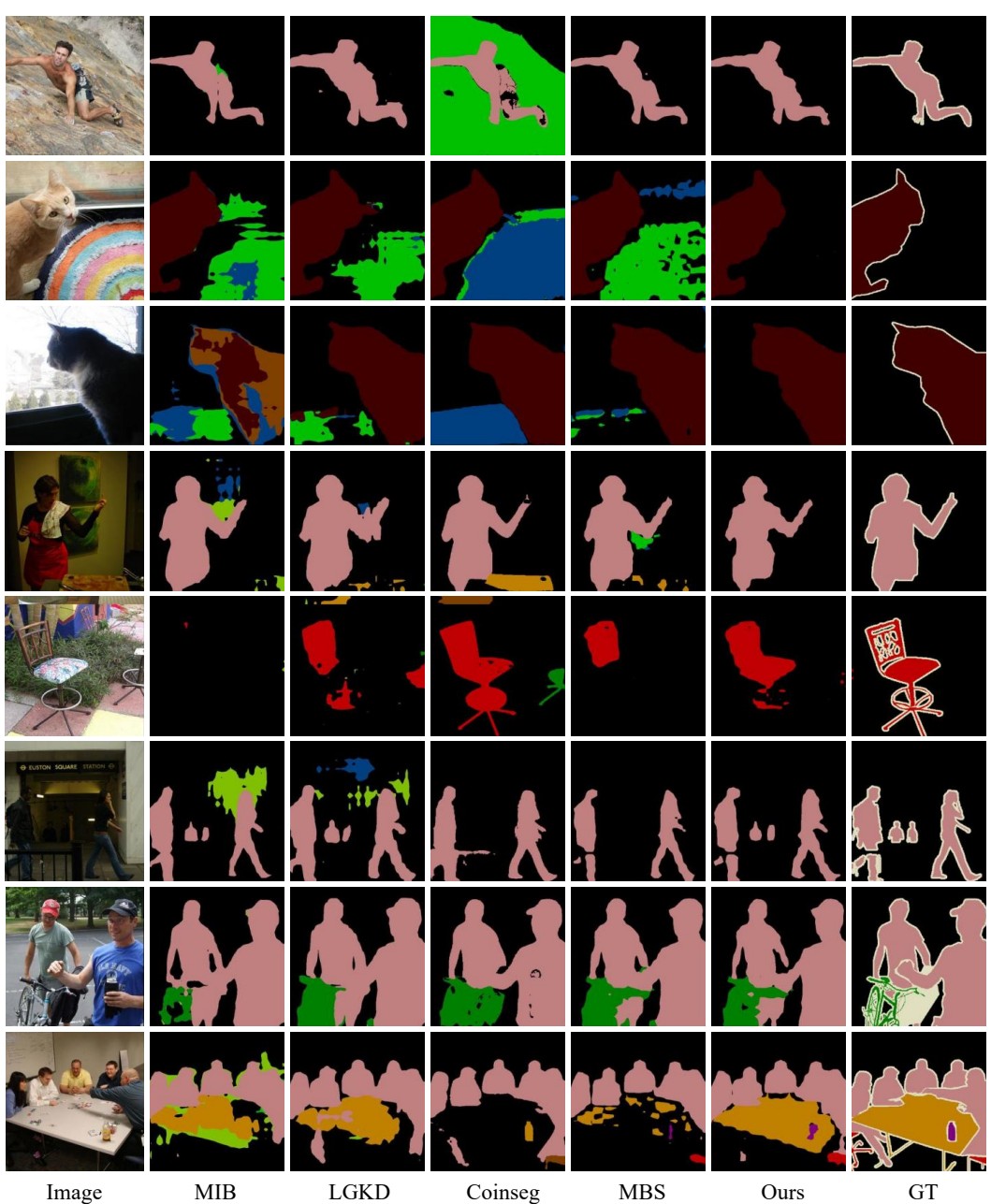

Figure 7: Qualitative analysis of results under 15-1 setting. Our method provides more accurate pixel-level segmentation for old classes with strong resistance to forgetting, while also reducing misclassification of new class pixels, demonstrating superior plasticity in learning new classes.

# E BROADER IMPACTS

In semantic segmentation, a common approach to handling newly emerging classes is to train the new and old class data together. However, due to storage limitations and data privacy concerns, old data is often inaccessible, rendering this approach impractical. Additionally, fine-tuning strategies are prone to catastrophic forgetting. In this paper, we propose a distribution-based incremental semantic segmentation learning strategy that mitigates forgetting without requiring access to old class data,

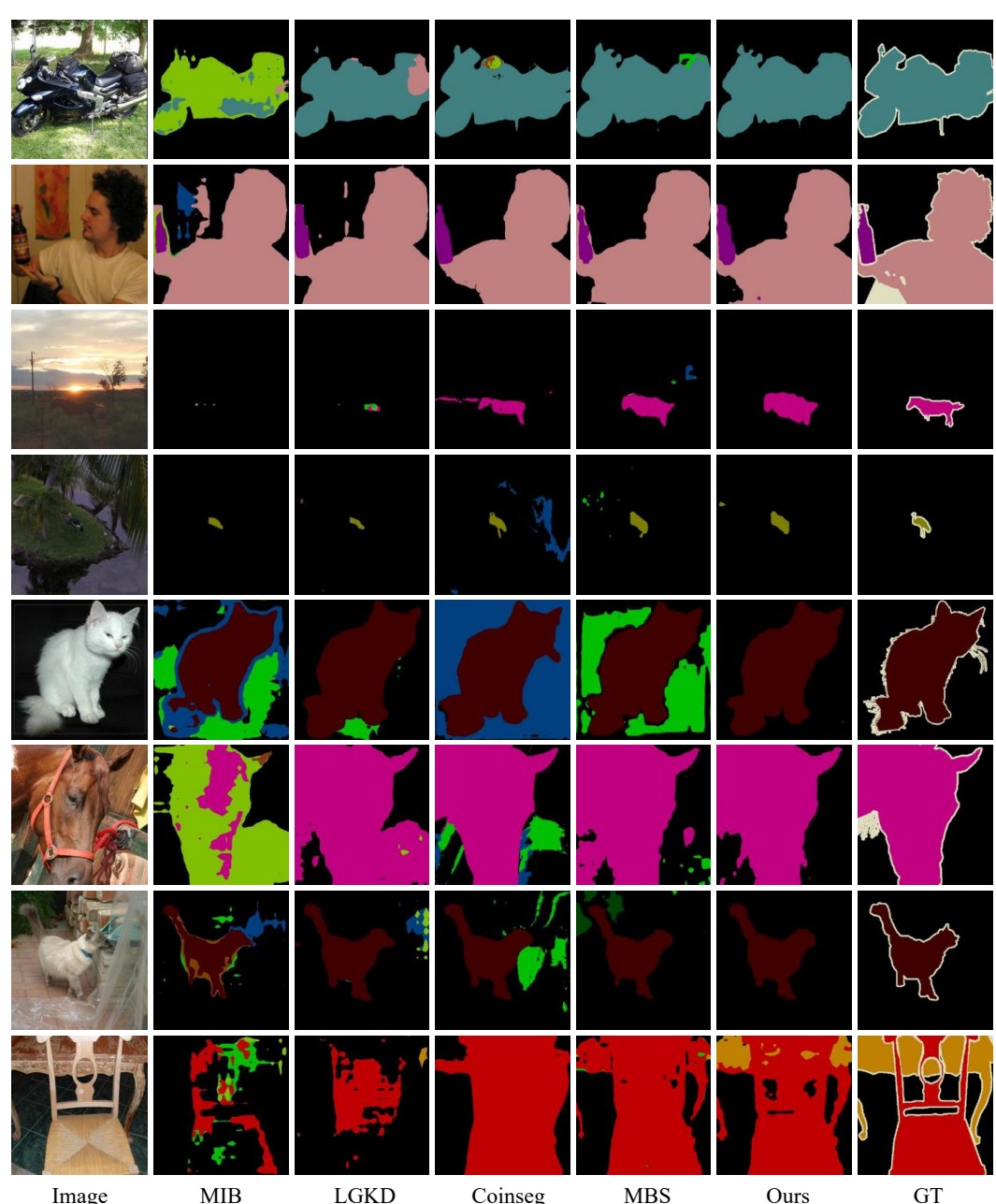

| Image | MIB | LGKD | Coinseg | MBS | Ours | GT |

Figure 8: Qualitative analysis of results under 15-1 setting. Our method provides more accurate pixel-level segmentation for old classes with strong resistance to forgetting, while also reducing misclassification of new class pixels, demonstrating superior plasticity in learning new classes.

while continuously learning new class knowledge. This method has promising applications in fields such as autonomous driving, medical image segmentation, and environmental monitoring.

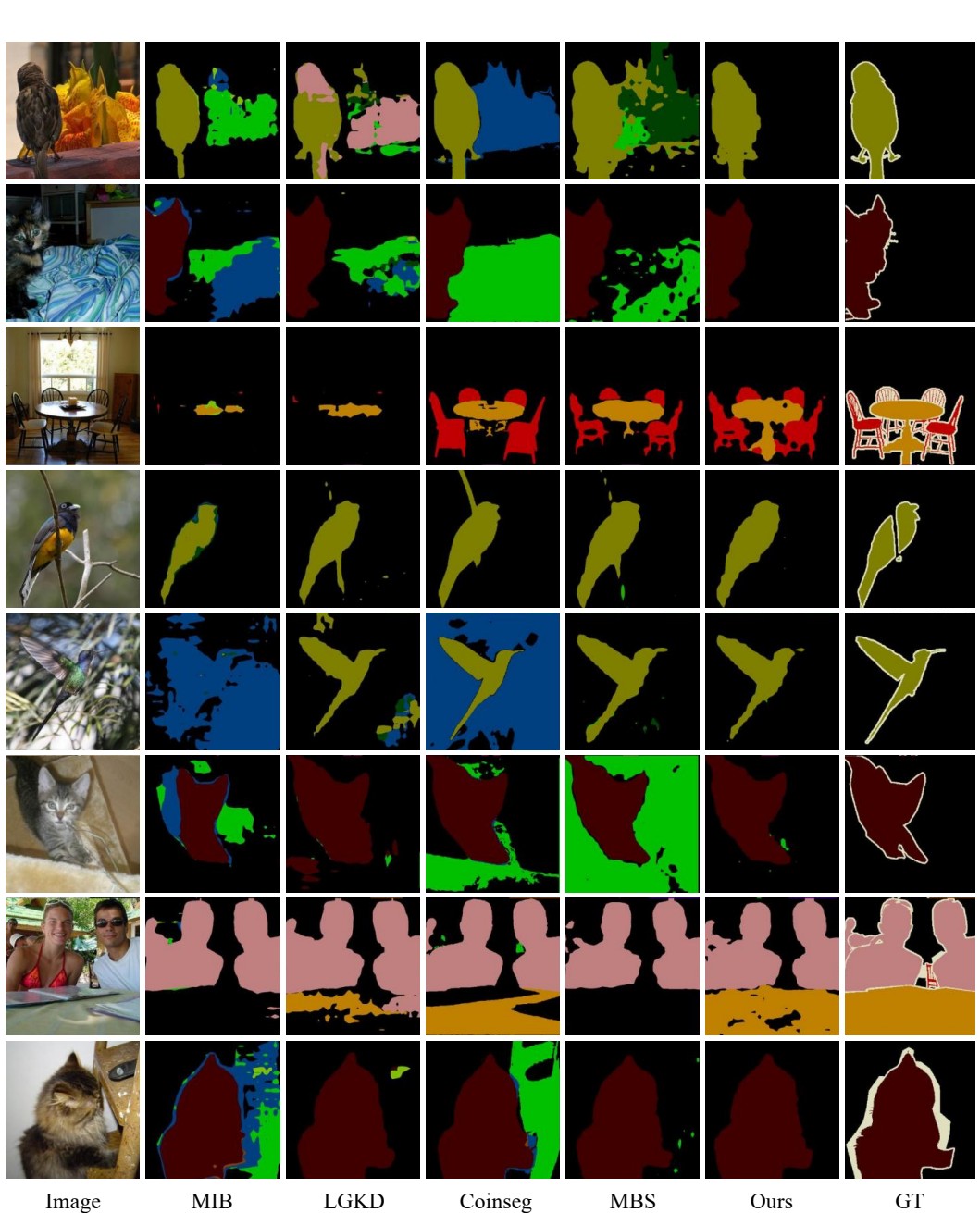

Figure 9: Qualitative analysis of results under 15-1 setting. Our method provides more accurate pixel-level segmentation for old classes with strong resistance to forgetting, while also reducing misclassification of new class pixels, demonstrating superior plasticity in learning new classes.

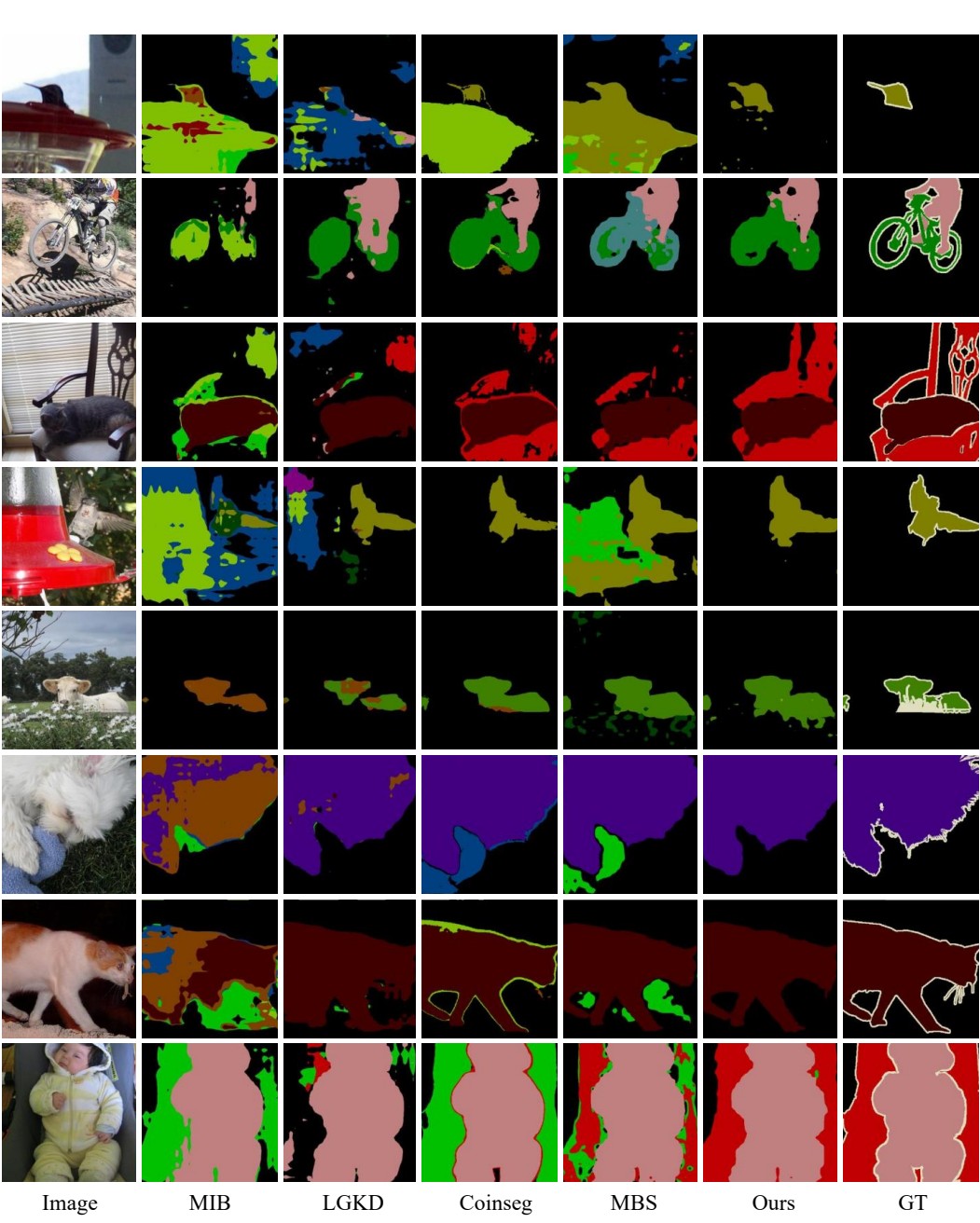

|         |     |      |         |     |      |    |
|---------|-----|------|---------|-----|------|-----|
| Image   | MIB | LGKD | Coinseg | MBS | Ours | GT |

Figure 10: Qualitative analysis of results under 15-1 setting. Our method provides more accurate pixel-level segmentation for old classes with strong resistance to forgetting, while also reducing misclassification of new class pixels, demonstrating superior plasticity in learning new classes.

