# OpenReview forum: "Parameter Release and Knowledge Reuse for Class-Incremental Semantic Segmentation"
_ICLR.cc/2026/Conference — ICLR 2026 Conference Withdrawn Submission_

### Official Review · Reviewer_dRri · 2025-10-27

**Soundness:** 2
**Presentation:** 2
**Contribution:** 1
**Rating:** 2
**Confidence:** 4

**Summary:**

This paper proposes a distribution-based knowledge distillation method to address catastrophic forgetting in class-incremental semantic segmentation (CISS) without storing past data. The authors argue that there is a parameter competition that standard knowledge distillation methods preserve old distributions too rigidly which limits parameter space for new classes and wasting previously acquired knowledge. They propose a min–max distribution strategy that releases low-sensitivity old parameters and minimizes the old-knowledge distribution used at the current step; estimates and reuses transferable old knowledge via a Laplacian-guided position map and a projection-based confidence map; and maximizes shared knowledge using an entropy-induced objective. In the proposed method, parameter is released by per-layer L2 norms with a pruning mask, then three losses are used for training. Experiments on standard datasets show that the proposed method has good performance across multiple incremental learning settings. I don't find the ideas and motivations to be really novel in continual learning or specific to semantic segmentation. The main contribution is mainly on implementation.

**Strengths:**

The loss objectives are convincingly defined from the motivations. Experiments show consistent gains on standard datasets across incremental learning settings. The training and inference overhead is low.

**Weaknesses:**

The ideas are not novel in continual learning or specific to semantic segmentation. It is hard to extract new insight from this paper into continual semantic segmentation problem.

**Questions:**

1. How sensitive are results to the pruning threshold, and to per-layer and per-stage thresholding?  Are masks recomputed every epoch/step?
2. What’s the added training wall-clock and peak memory overhead of the Laplacian and projection maps at 512 x 512? Do these costs scale linearly with resolution?

---

### Official Review · Reviewer_vbrR · 2025-10-30

**Soundness:** 3
**Presentation:** 3
**Contribution:** 2
**Rating:** 4
**Confidence:** 4

**Summary:**

This paper explores the challenges of continuous semantic segmentation by proposing a method called Distribution-based Distillation. Specifically, the authors designed three different additional loss functions to aid incremental semantic segmentation and utilized laplacian-based projection to generate two maps to assist learning. Experiments on multiple datasets demonstrate its effectiveness.

**Strengths:**

1. The experimental results are very good, and the proposed distribution-based distillation is relatively novel to a certain extent.

**Weaknesses:**

1. The authors propose a layer-wise pruning in order to refine the previous model. How did the authors choose the pruning threshold? Plus, is there a comparison with PackNet[1] here and an explanation of the benefits?

[1] PackNet: Adding Multiple Tasks to a Single Network by Iterative Pruning, CVPR 2018

**Questions:**

See the Weaknesses Above.

---

### Official Review · Reviewer_koAs · 2025-10-31

**Soundness:** 3
**Presentation:** 3
**Contribution:** 3
**Rating:** 4
**Confidence:** 4

**Summary:**

This paper proposes an innovative approach that combines the Parameter Release (PR) and Knowledge Reuse (KR) mechanisms to address the critical challenge of catastrophic forgetting in Class-Incremental Semantic Segmentation (CISS). The paper's motivation is clear, effectively highlighting the limitations of overly strict constraints imposed by traditional Knowledge Distillation (KD), and suggesting a solution that selectively releases model capacity to allocate space for new class learning, supplemented by the KR module to consolidate both old and new knowledge.

**Strengths:**

The paper's motivation is clear, effectively highlighting the limitations of overly strict constraints imposed by traditional Knowledge Distillation (KD), and suggesting a solution that selectively releases model capacity to allocate space for new class learning, supplemented by the KR module to consolidate both old and new knowledge.

**Weaknesses:**

‌Although DKD alleviates the competition between old and new knowledge through parameter release, it fundamentally still relies on soft target alignment from the old model. When the old model itself contains errors or biases, these issues may propagate to the new model through the distillation process. Some methods related to large models need to be considered for comparison.

**Questions:**

1 Eq11 contains 4 loss terms, does it require parameters to control their weights?
2 What is the correlation and correspondence between theoretical analysis and experimental parts?

---

### Note · Authors · 2025-11-18

I have read and agree with the venue's withdrawal policy on behalf of myself and my co-authors.